# Histological hallmarks and role of Slug/PIP axis in pulmonary hypertension secondary to pulmonary fibrosis

Gregoire Ruffenach[1], Soban Umar[1], Mylene Vaillancourt[1], Jason Hong[1,2], Nancy Cao[1], Shervin Sarji[1], Shayan Moazeni[1], Christine M Cunningham[1], Abbas Ardehali[3], Srinivasa T Reddy[4], Rajan Saggar[2], Gregory Fishbein[5] & Mansoureh Eghbali[1,*] (ID)

## Abstract

Pulmonary hypertension secondary to pulmonary fibrosis (PF-PH) is one of the most common causes of PH, and there is no approved therapy. The molecular signature of PF-PH and underlying mechanism of why pulmonary hypertension (PH) develops in PF patients remains understudied and poorly understood. We observed significantly increased vascular wall thickness in both fibrotic and non-fibrotic areas of PF-PH patient lungs compared to PF patients. The increased vascular wall thickness in PF-PH patients is concomitant with a significantly increased expression of the transcription factor Slug within the macrophages and its target prolactin-induced protein (PIP), an extracellular matrix protein that induces pulmonary arterial smooth muscle cell proliferation. We developed a novel translational rat model of combined PF-PH that is reproducible and shares similar histological features (fibrosis, pulmonary vascular remodeling) and molecular features (Slug and PIP upregulation) with human PF-PH. We found Slug inhibition decreases PH severity in our animal model of PF-PH. Our study highlights the role of Slug/PIP axis in PF-PH.

**Keywords** non-fibrotic area; prolactin-induced protein; Slug; vascular remodeling
**Subject Category** Respiratory System

## Introduction

Pulmonary hypertension (PH) is a life-threatening disease characterized by an increased mean pulmonary arterial pressure (mPAP) over 25 mmHg. This increased pressure is due, at least in part, to the thickening of distal pulmonary arterial walls as a result of uncontrolled proliferation of vascular wall cells (Vaillancourt *et al*, 2015; Sharma *et al*, 2016). One of the most common forms of PH is secondary to interstitial lung disease (group 3.2) with a prevalence of about 30–40% in patients with pulmonary fibrosis (PF) (Lettieri *et al*, 2006; Behr & Ryu, 2008; Simonneau *et al*, 2013). PH secondary to PF is refractory to conventional drug therapies and is a strong indication for lung transplantation. Despite the high prevalence of PH in this population, the molecular mechanism of PH secondary to PF (PF-PH) remains under-investigated (Patel *et al*, 2007). PF-PH has been viewed as a process driven by lung fibrosis. This view is supported by the study from Colombat *et al* (2007) who found a significant positive correlation between the macroscopic extent of lung fibrosis and mean pulmonary artery pressure in idiopathic PF patients, and by other studies reporting a correlation between vascular wall thickness and severity of lung fibrosis in PF (Parra *et al*, 2005; Mlika *et al*, 2015). However, other clinical reports did not find any correlation between increased mPAP and lung fibrosis (Mura *et al*, 2012) as measured by high-resolution chest CT or forced vital capacity of PF patients (Nadrous *et al*, 2005; Zisman *et al*, 2007; Nathan *et al*, 2019). Furthermore, a growing body of evidence (Heath *et al*, 1968; Harari *et al*, 1997; Leuchte *et al*, 2004; Kim *et al*, 2010) demonstrates that molecular differences exist between PF and PF-PH patients, suggesting the involvement of specific pathways in PF-PH patients. Pathological reports suggest that vascular remodeling in PF patients is mainly limited to fibrotic areas of the lung, whereas in PF-PH patients, it spreads to the non-fibrotic area as well, although this histological observation was never quantified (Heath *et al*, 1968; Kim *et al*, 2010). Taken together, these reports further support a specific molecular mechanism of PH development in PF-PH patients.

At the molecular level, the transcription factor Slug, also known as Snai2, has already been implicated in PF (Jayachandran *et al*, 2009) where it is expressed by epithelial cells and can participate in

1 Division of Molecular Medicine, Department of Anesthesiology & Perioperative Medicine, UCLA, Los Angeles, CA, USA
2 Division of Pulmonary and Critical Care, Department of Medicine, UCLA, Los Angeles, CA, USA
3 Division of Cardiothoracic Surgery, Department of Surgery, UCLA, Los Angeles, CA, USA
4 Division of Molecular & Medical Pharmacology, Department of Medicine, UCLA, Los Angeles, CA, USA
5 Department of Pathology, UCLA, Los Angeles, CA, USA
*Corresponding author. Tel: +310 206 0345; E-mail: meghbali@ucla.edu

extracellular matrix remodeling (Zhang *et al*, 2011; Boufraqech *et al*, 2016; Coll-Bonfill *et al*, 2016). Slug has also been implicated in chronic obstructive pulmonary disease (Hopper *et al*, 2016) as well as in pulmonary arterial hypertension (Ranchoux *et al*, 2015; Tomos *et al*, 2017), where it is expressed by endothelial cells and participates in endothelial to mesenchymal transition. Nonetheless, the role of Slug has never been investigated in PF-PH patients to date.

In the present study, we have carefully examined the histological features of the pulmonary vasculature between non-fibrotic and fibrotic areas of the lung in PF and PF-PH patients. We also elucidated a molecular signature of PF-PH that could drive the histological differences in non-fibrotic areas between PF-PH and PF lungs. We found that the transcription factor Slug is upregulated in PF-PH when compared to PF lungs. In PF-PH patient lungs, we found that upregulation of Slug was associated with increased vascular cell proliferation via its target the prolactin-induced protein (PIP). We developed a novel translational rat model of combined PF-PH that is reproducible and shares similar histological features (fibrosis, pulmonary vascular remodeling) and molecular features (Slug and PIP upregulation) with human PF-PH. Our data show that Slug is expressed in epithelial cells, macrophages, and fibroblasts, but Slug expression is only significantly higher in macrophages in PF-PH compared to PF both in human and in rat lungs. We also demonstrate that Slug inhibition in our pre-clinical model of PF-PH decreases PH severity and is associated with reduced PIP expression. Our study reveals a new role for Slug in promoting vascular remodeling through its extracellular target PIP.

## Results

### Vascular wall thickening in non-fibrotic areas of distal pulmonary arteries in patients with PH secondary to PF

We first examined differences in lung fibrosis and vascular remodeling between patients with PF-PH and PF alone. The Ashcroft score between PF and PF-PH patients was similar, but significantly higher compared to Ctrl and PAH patients (Fig 1A). mRNA expression of fibronectin, collagen I, and collagen III was also not significantly different between PF and PF-PH patients (Fig 1B–D). Furthermore, vascular wall thickness was significantly increased in PAH and PF-PH patients compared to Ctrl and PF. PF patients also exhibited increased vascular thickening compared to Ctrl, but it was significantly lower than in PF-PH patients (Fig 1E).

To examine whether there are any differences in vascular remodeling in areas of lung with different degree of fibrosis, vascular remodeling was quantified in three different areas: non-fibrotic (Ashcroft score 0–2), mild fibrosis (Ashcroft score 2–5), and severe fibrosis (Ashcroft score > 5) in PF and PF-PH patients. There was a direct correlation between Ashcroft score and distal vascular wall thickness in our end-stage PF group. We observed the same correlation in the PF-PH patient population but with a significantly higher degree of vascular remodeling (Fig 1F). We then grouped mild and severe fibrotic areas together to investigate vascular wall thickness in non-fibrotic (Ashcroft score < 2) and fibrotic areas (Ashcroft score > 2) in PF-PH versus PF patients. In non-fibrotic areas, significant pulmonary vascular thickening was found in PF-PH versus PF patients (Fig 1G). In both PF and PF-PH patients, fibrotic areas yielded increased vascular remodeling when compared to non-fibrotic areas of the lung (Fig 1G). Taken together, our data show increased vascular wall thickening in the non-fibrotic and fibrotic areas of PF-PH patients compared to PF, demonstrating a distinct vascular remodeling pattern between PF-PH and PF patients.

### Increased proliferation of vascular wall cells in PF-PH patients is associated with upregulation of Slug and its target PIP

We investigated cell proliferation within the distal pulmonary vascular walls, a mechanism well known to participate in vascular wall thickening in PH. Pulmonary vascular endothelial cells (EC) and pulmonary vascular smooth muscle cells (SMC) were more proliferative in PF-PH compared to PF patients assessed by the percent of Ki67-positive nuclei (Fig 2A). Given the known implication of the transcription factor Slug in cell proliferation (Emadi Baygi *et al*, 2010; Yang *et al*, 2010) and extracellular matrix remodeling (Huang *et al*, 2009), we compared the expression of Slug in the lungs of PF-PH and PF patients. We found a 2.5-fold upregulation of Slug mRNA and protein in PF-PH lungs compared to PF alone (Fig 2B and C). In order to define which cell type(s) is responsible for Slug upregulation in the lungs of PF-PH patients, we measured the expression of Slug in 5 major cell types in the lungs: epithelial cells, fibroblasts, macrophages, EC, and SMC. Our co-immunofluorescence staining showed Slug is highly expressed in the epithelial cells and to lesser extent in macrophages, fibroblasts, and EC (Fig 2D). More importantly, we found that expression of Slug is significantly increased only in macrophages, without any change in CD68 transcript levels, in PF-PH compared to PF patients (Fig 2E). Furthermore, Slug expression was more abundant in the fibrotic areas of the lung in

**Figure 1. Pulmonary vascular remodeling pattern in PF-PH patients is distinct from patients with PF alone.**

A    Masson's trichrome representative images of pulmonary fibrosis and quantification.

B–D    Relative transcript expression of (B) fibronectin, (C) collagen I, and (D) collagen III to GAPDH.

E    Masson's trichrome representative images of vascular wall thickening and quantification.

F    Masson's trichrome representative images and correlation between lung fibrosis and vascular wall thickness in PF and PF-PH patients. Out of the 14 PF and PF-PH patients, 3 of them had no non-fibrotic area and therefore were excluded for comparison between non-fibrotic and fibrotic areas (*n* = 11/group; Ashcroft score non-fibrosis: 0–2; mild fibrosis: 3–5; severe fibrosis > 5) (* compared to non-fibrotic areas of the same group of patients).

G    Masson's trichrome representative images of vascular wall thickening in non-fibrotic area (Ashcroft score 0–2) and fibrotic area (Ashcroft score > 2) and quantification in PF and PF-PH.

Data information: Values are expressed as mean ± SEM. The number of samples per group for each experiment is written within each bar graph. Statistical test: panels (A, E–G): ANOVA; panels (B–D): *t*-test (* versus Ctrl, † versus PF, ! versus PAH, $ versus PF-PH; **$P < 0.01$, ***$P < 0.001$, ****$P < 0.0001$, $^{††††}P < 0.0001$, $^{!!!}P = 0.0001$, $^{!!!!}P < 0.0001$, $^{$$$$}P < 0.0001$).

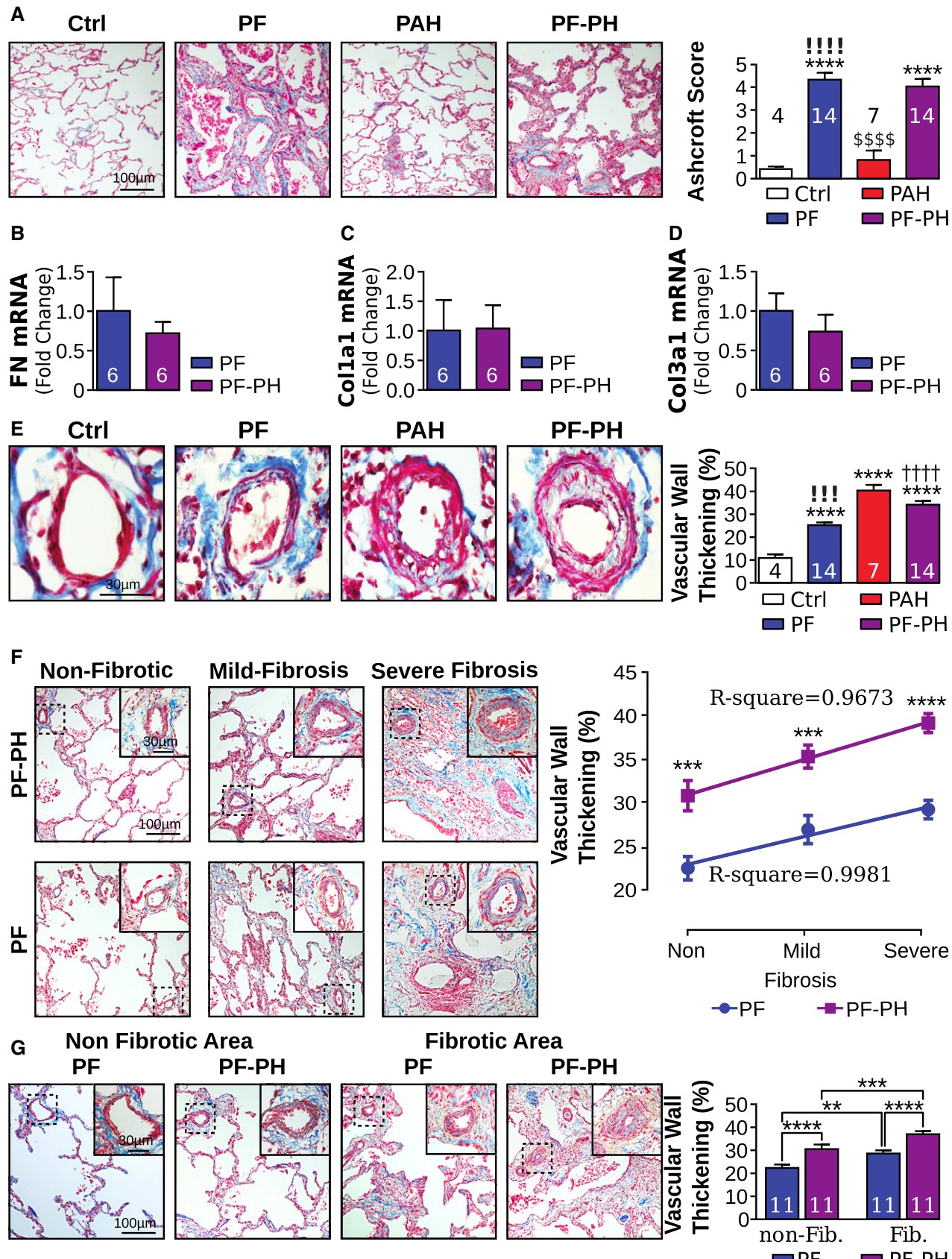

**Figure 1.**

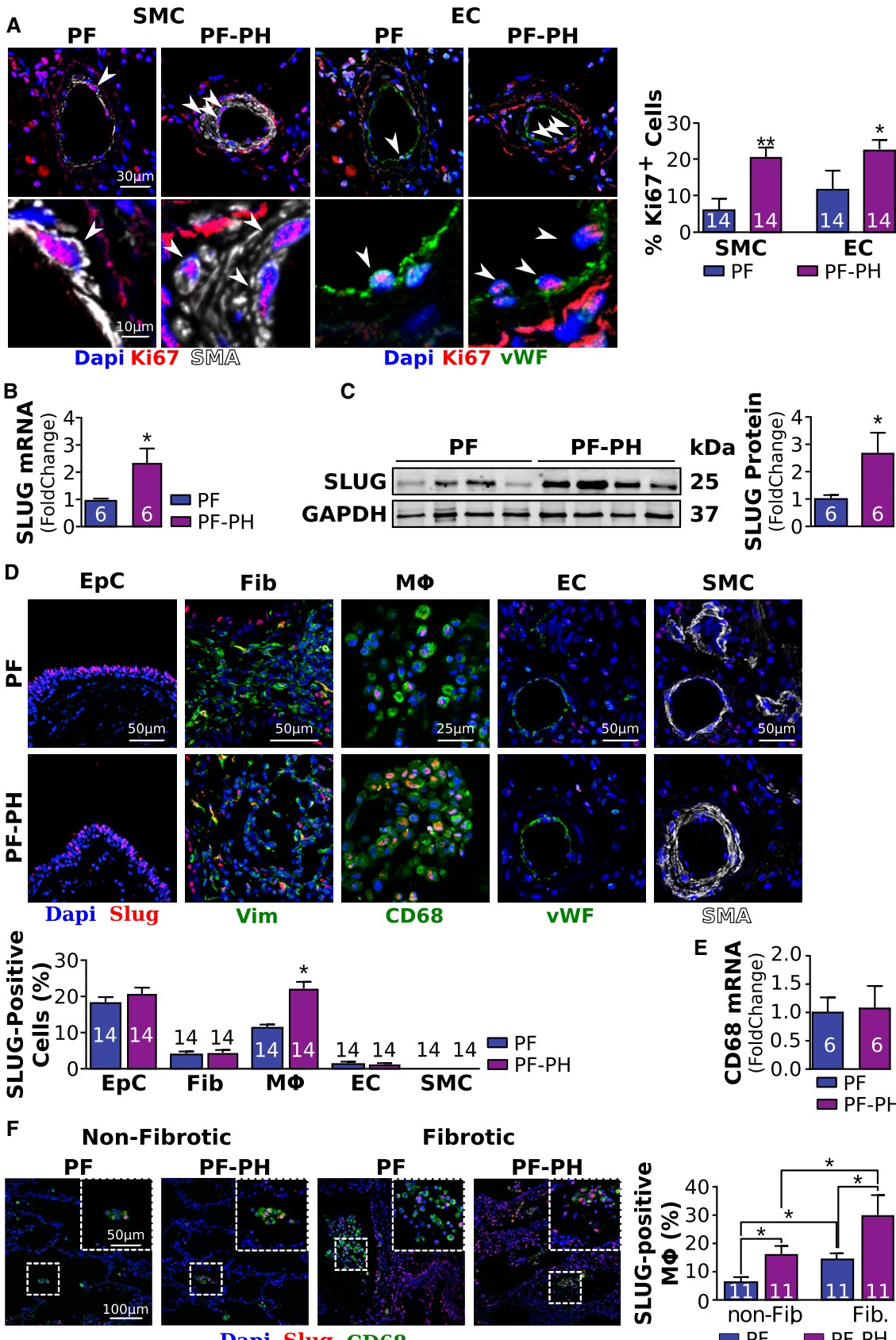

Figure 2.

**Figure 2.  PF-PH patients exhibit increased proliferation of vascular wall cells and increased Slug expression.**

A       Representative images of Ki67-positive pulmonary vascular endothelial cells (EC, vWF in green) and smooth muscle cells (SMC, aSMA in white) and quantification (* versus PF, *P < 0.05, **P < 0.01).

B, C    Relative expression of Slug mRNA (B) and protein (C) normalized to GAPDH in PF and PF-PH patients.

D       Representative images of Slug (red) expression in bronchial epithelial cells (EpC), fibroblasts (Fib) (vimentin in green), macrophages (Mϕ, CD68 in green), pulmonary vascular endothelial cells (EC, von Willebrand factor in green), and pulmonary vascular smooth muscle cells (SMC, aSMA in white); and quantification.

E       Relative CD68 mRNA expression normalized to GAPDH.

F       Representative images of Slug (red) expression in macrophages (Mϕ, CD68 in green) in non-fibrotic and fibrotic areas of the lung in PF and PF-PH patients and quantification.

Data information: Values are expressed as mean ± SEM. The number of samples per group for each experiment is included within each bar graph. Statistical test: panels (A–E): t-test; panel (F): ANOVA (* versus PF; *P < 0.05).

both PF-PH and PF patients when compared to the non-fibrotic areas (Fig 2F).

Based on our data, we hypothesized that Slug might be responsible for promoting vascular cell proliferation in PF-PH patients by regulating the expression of extracellular matrix proteins. To test this hypothesis, we used publicly available microarray data from 22 PF and 45 PF-PH patients (Mura et al, 2012). The clinical characteristics of the patients used in the microarray were comparable with our cohort (Table 2 from Mura et al, 2012). Our analysis revealed a significant upregulation of 59 genes known to be transcriptional targets of Slug in PF-PH patients compared to PF (Fig 3A, and Appendix Table S5). We found seven of these genes are implicated in cellular proliferation and present in the extracellular space (Fig 3B and C, Appendix Tables S6 and S7). Since the most upregulated gene, mucin 4, is anchored to the cell membrane (Dhanisha et al, 2018) and is not secreted into the extracellular matrix, we focused on the second most upregulated transcriptional target of Slug, the prolactin-induced protein (PIP). We observed that PIP mRNA and protein were upregulated in the lungs of our PF-PH patient population compared to the PF population (Fig 3D and E). Immuno-staining confirmed PIP upregulation in the extracellular space (Fig 3F).

To demonstrate the functional role of PIP on vascular cells, we exposed human fibroblasts, EC, and SMCs to PIP and assessed cell proliferation. We found PIP did not induce fibroblast proliferation (Fig 3G) but was sufficient to induce EC and SMC proliferation (Fig 3H–J).

Taken together, Slug upregulation in the lungs of PF-PH patients is associated with increased expression of its transcriptional target PIP leading to vascular cell proliferation.

## Development of a new animal model recapitulating the histological hallmarks of PH secondary to PF

Based on our above findings, we hypothesized that Slug inhibition may reduce the severity of PH in PF-PH by targeting PIP. Bleomycin has been used for decades to induce PF; however, the vascular remodeling in bleomycin-treated rats has been shown to be only present in fibrotic areas of the lung (Jarman et al, 2014). We also found rats treated with bleomycin do not recapitulate histological and molecular features of our PF-PH patients and were, therefore, not suitable to test our hypothesis (Figs 4 and 5).

To circumnavigate this, we developed an animal model of PH in pre-existing PF, by sequential instillation of bleomycin and MCT. Since bleomycin treatment is known to produce two distinct phases, an early inflammatory phase and a fibrotic phase starting around day nine (Chaudhary et al, 2006), we started MCT after the second phase to ensure PH development in a pre-existing fibrotic environment similar to human pathology. Our data at day 14 post-bleomycin show a significant increase of the Ashcroft score, as well as the mRNA expression of fibronectin, collagen I, and collagen III compared to Ctrl, confirming the development of pulmonary fibrosis in these animals at this time point (Fig 4A–D). Therefore, MCT was administered 14 days post-bleomycin instillation, and all rats were sacrificed at day 35 as shown in Fig 4E. Measurement of RVSP and RV hypertrophy index confirmed the development of PH in this model of PF-PH. Indeed, the RVSP of Bleo-MCT and MCT rats were similar but significantly higher than Ctrl and Bleo rats (Fig 4F and G).

To confirm the relevance of this model to human disease, we assessed lung fibrosis and vascular remodeling. As in humans, we found the Ashcroft score and mRNA expression of collagen I and collagen III were similar in Bleo and Bleo-MCT rats and significantly higher when compared to the Ctrl and MCT groups (Fig 4H–K). The expression of fibronectin was similar between MCT, Bleo, and Bleo-MCT rats but significantly higher than Ctrl (Fig 4I). Furthermore, vascular wall thickness was similar and increased in MCT and MCT-Bleo rats when compared to Ctrl and Bleo rats (Fig 4L).

The comparison between fibrotic and non-fibrotic areas of the lung showed that Bleo rats had significantly thicker vascular walls in fibrotic areas compared to non-fibrotic areas (Fig 4M). Furthermore, Bleo-MCT rats showed a significantly increased distal

**Figure 3.  Online lung microarray analysis revealed PIP as one of the Slug transcriptional targets mediating vascular cell proliferation in PF-PH.**

A       Heatmap of the 20 genes upregulated in GSE24988 microarray and known to be Slug transcriptional target.

B       Venn diagram showing the overlap between Slug targets implicated in cell proliferation and known to be extracellular.

C       Heatmap of the seven transcriptional targets of Slug known to promote proliferation and to be extracellular.

D, E    Relative expression of prolactin-induced protein (PIP) mRNA (D) and protein (E) expression normalized to GAPDH in PF and PF-PH patients.

F       Representative images of PIP immunohistochemistry showing its extracellular localization and quantification.

G–J     Proliferation assay on healthy fibroblast (G), pulmonary arterial EC (H), and healthy pulmonary arterial SMC (I and J) in the presence or absence of PIP.

Data information: Values are expressed as mean ± SEM. The number of samples per group for each experiment is included within each bar graph. Statistical test: panels (D–F): t-test; panels (G–J): ANOVA (*P < 0.05, **P < 0.01 ***P < 0.001).

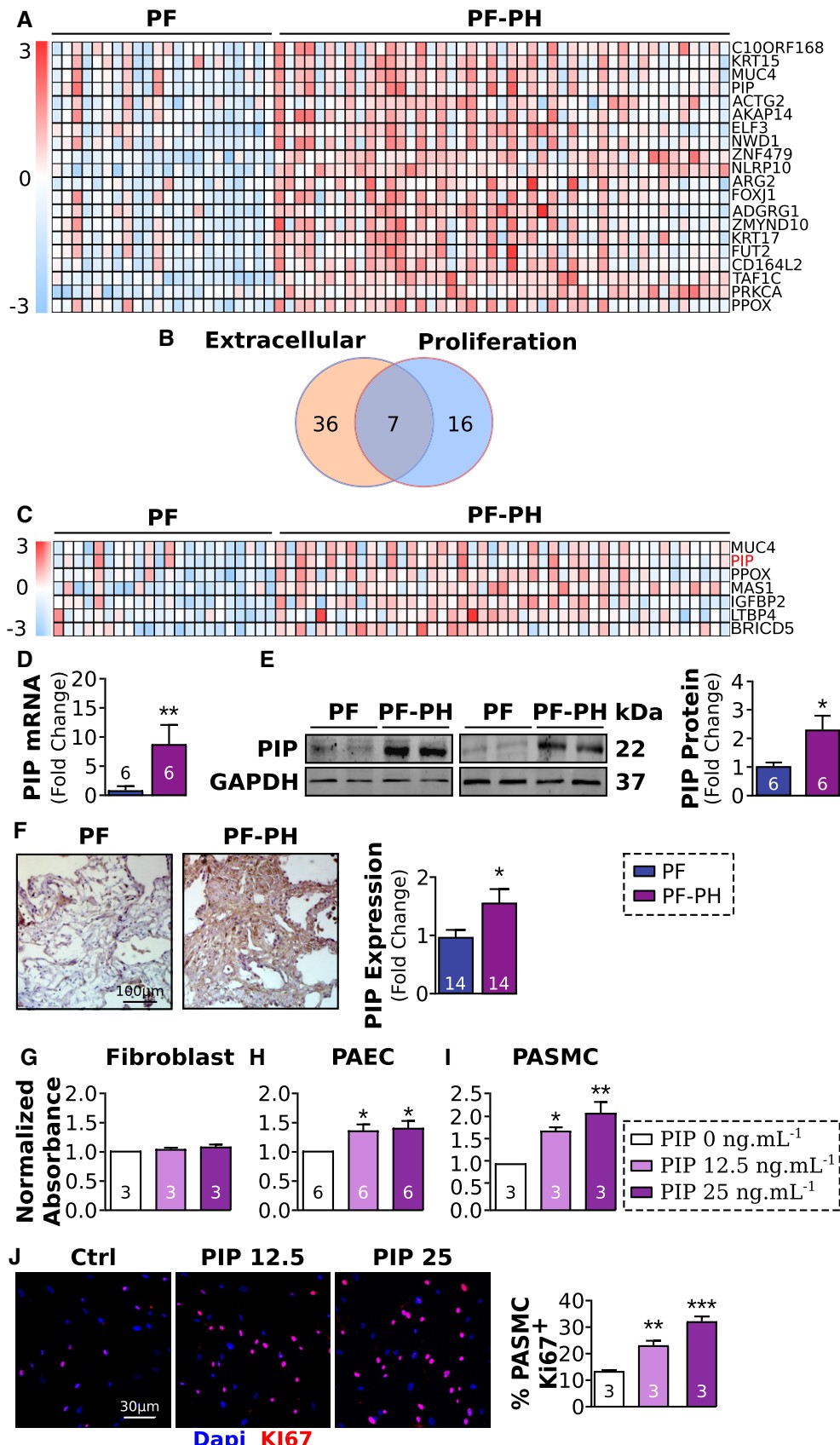

Figure 3.

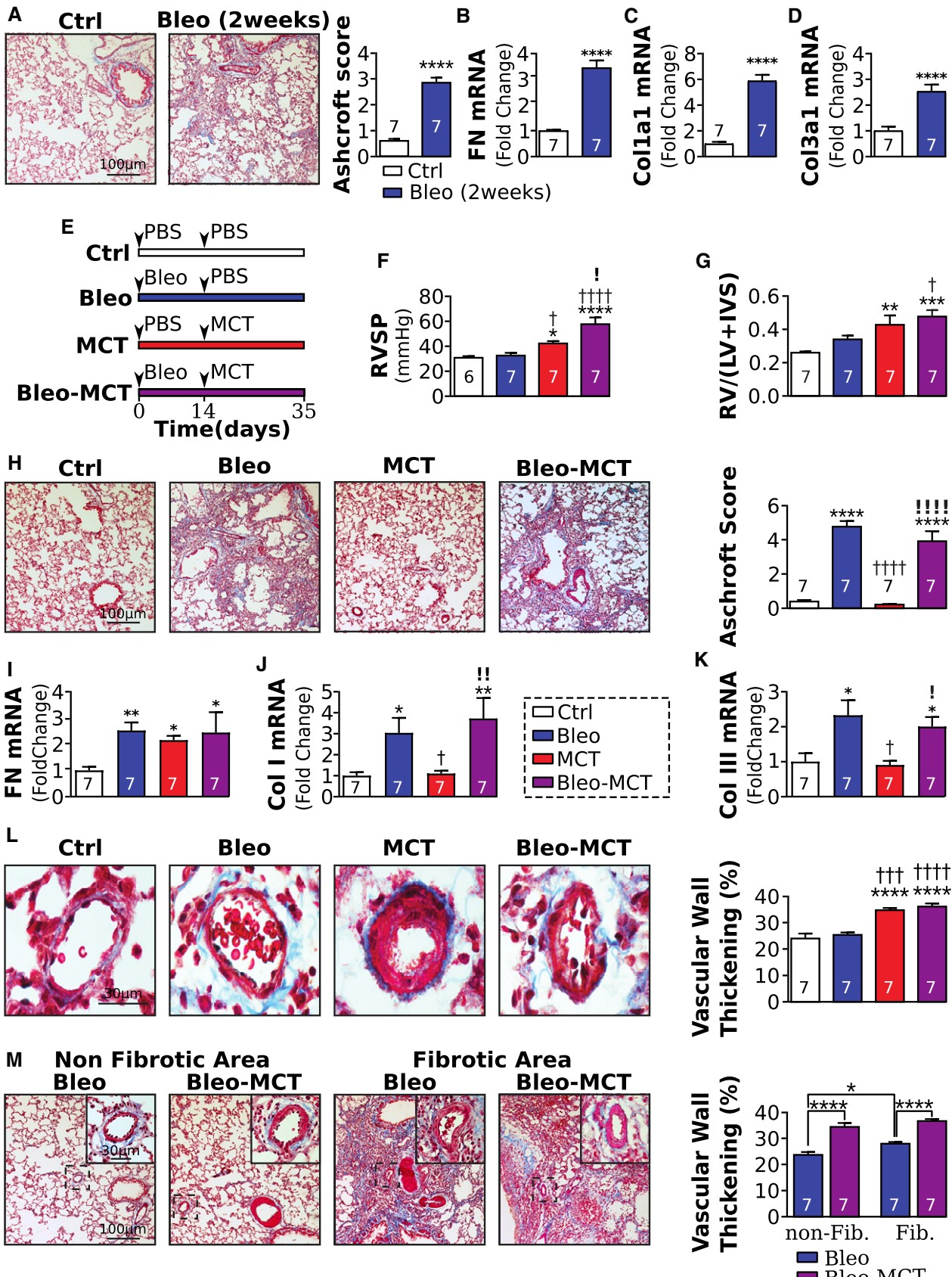

**Figure 4.**

**Figure 4.  An animal model recapitulating most of the histological hallmarks of PH secondary to PF.**

A       Masson's trichrome representative images of lung fibrosis and quantification 2 weeks after PBS or bleomycin instillation.

B–D    Quantification of (B) fibronectin, (C) collagen I, and (D) collagen III mRNA.

E       Schematic of experimental protocol. Animals were divided into 4 groups: control (Ctrl), pulmonary fibrosis (PF), pulmonary hypertension (PH), and PF secondary to
        PH (PF-PH). At day 0, rats in the PF and PF-PH groups received an intratracheal instillation of bleomycin (2.5 mg/kg), whereas the Ctrl and PH groups received a
        PBS instillation. At day 14, the PH and PF-PH groups received a subcutaneous injection of monocrotaline (MCT, 60 mg/kg), whereas the Ctrl and PF groups received
        a subcutaneous injection of PBS.

F, G    (F) Right ventricular systolic pressure (RVSP) and (G) right ventricular hypertrophy index following the experimental protocol.

H       Masson's trichrome representative images and quantification of Ashcroft score in the experimental model.

I–K     Relative expression of fibronectin (I), collagen I (J), and collagen III (K) mRNA normalized to GAPDH.

L       Masson's trichrome representative images and quantification of vascular remodeling.

M       Masson's trichrome representative images and quantification of the vascular remodeling between fibrotic and non-fibrotic areas in the experimental model.

Data information: Values are expressed as mean $\pm$ SEM. The number of samples per group for each experiment is included within each bar graph. Statistical test: panels
(A–D) : $t$-test; panels (F–M): ANOVA (* versus Ctrl, † versus PF, ! versus PH; *$P < 0.05$, **$P < 0.01$, ***$P < 0.001$, ****$P < 0.0001$, †$P < 0.05$, †††$P < 0.001$. ††††$P < 0.0001$,
!$P < 0.05$, !!$P < 0.01$, !!!!$P < 0.0001$).

vascular wall thickness in fibrotic and non-fibrotic areas of the lung compared to Bleo rats, consistent with our observations in the human disease (Fig 4M).

Collectively, these data provide the first evidence that our animal model fulfills distal pulmonary vascular histological characteristics of PH in end-stage PF-PH patients, while bleomycin alone is not sufficient to recapitulate these histological features of PF-PH patients.

**Our animal model of PF-PH exhibits a pattern of lung vascular wall cell proliferation and upregulation of Slug/PIP similar to patients**

We further compared lung vascular wall cell proliferation in our animal model of PF-PH to human PF-PH pathology and to the rat model of PF. Similar to patients, MCT-Bleo rats had a significantly higher percent of Ki67-positive nuclei in EC and SMC in comparison with Bleo rats (Fig 5A).

Next, we examined the expression of Slug and PIP in our animal model. We found the expression of Slug mRNA and protein was significantly upregulated in the lungs of Bleo-MCT rat lungs compared to Bleo alone (Fig 5B and C). In order to define which cell type(s) is responsible for Slug upregulation in the lung of Bleo-MCT rats, we measured the expression of Slug in epithelial cells, fibroblasts, macrophages, EC, and SMC. We found varying levels of Slug expression in epithelial cells, fibroblasts, and macrophages, but only in macrophages was the expression of Slug upregulated in Bleo-MCT compared to Bleo rats similar to humans (Fig 5D), without significant modification in CD68 mRNA (Fig 5E). Furthermore, in Bleo-MCT rats, both fibrotic and non-fibrotic areas had greater

expression of Slug compared to Bleo alone, although the expression of Slug in Bleo-MCT rats was significantly higher in fibrotic areas when compared to non-fibrotic areas (Fig 5F).

Finally, the expression of PIP mRNA and protein levels was also significantly increased in the lungs of Bleo-MCT rats compared to Bleo rats (Fig 6A–C).

Taken together, these data demonstrate that our combined animal model of PF-PH mimics the vascular remodeling and Slug and PIP expression of human end-stage PF-PH patients, whereas the bleomycin model of PF alone does not. Thus, our animal model allowed us to examine the potential of Slug as a therapeutic target to decreased PH severity in PF-PH.

**Slug inhibition decreases PH severity in rats with pre-existing PF and is associated with decreased PIP expression**

To test our hypothesis that Slug inhibition reduces severity of PH in PF-PH by targeting PIP, we silenced Slug in the lungs of Bleo-MCT rats (Fig 7A). Our data show Si-Slug resulted in significantly decreased Slug mRNA and protein expression compared to rats that received scramble siRNA (Fig 7B and C), demonstrating the efficiency of our knockdown. Slug inhibition in the presence of a PH stimulus was sufficient to reduce PH severity in rats with pre-existing PF as RVSP and right ventricular hypertrophy were significantly lower compared to scramble siRNA (Si-Scrm)-treated rats (Fig 7D and E).

Slug inhibition did not affect lung fibrosis since mRNA expression of fibronectin, collagen I, and collagen III as well as Ashcroft scores was similar between Si-Slug and Si-Scrm rats (Fig 7F–I). Nonetheless, vascular wall thickness was significantly lower in

**Figure 5.  Our animal model of PF-PH exhibits similar vascular wall cell proliferation and Slug expression to PF-PH patients.**

A       Representative images and quantification of Ki67-positive (red) pulmonary vascular endothelial cells (EC, vWF in green) and smooth muscle cells (SMC) in the
        experimental model (* versus Bleo, **$P < 0.01$).

B, C    Relative expression of Slug mRNA (B) and protein (C) normalized to GAPDH.

D       Representative images of Slug (red) expression in bronchial epithelial cells (EpC), fibroblasts (Fib, vimentin in green), macrophages (Mϕ, CD68 in green), pulmonary
        vascular endothelial cells (EC, vWF), and smooth muscle cells (SMC, aSMA in white); and quantification (below) (* versus Bleo, $P < 0.05$).

E       CD68 mRNA quantification.

F       Representative images of Slug (red) expression in macrophages (Mϕ, CD68 in green) in non-fibrotic and fibrotic areas of the lung in Bleo and Bleo-MCT rats and
        quantification.

Data information: Values are expressed as mean $\pm$ SEM. The number of samples per group for each experiment is included within each bar graph. Statistical test: panels
(A, D, and E): $t$-test; panels (B, C, and F): ANOVA (* versus Ctrl, † versus PF, ! versus PH; *$P < 0.05$, **$P < 0.01$, †$P < 0.05$, !$P < 0.05$, !!$P < 0.01$).

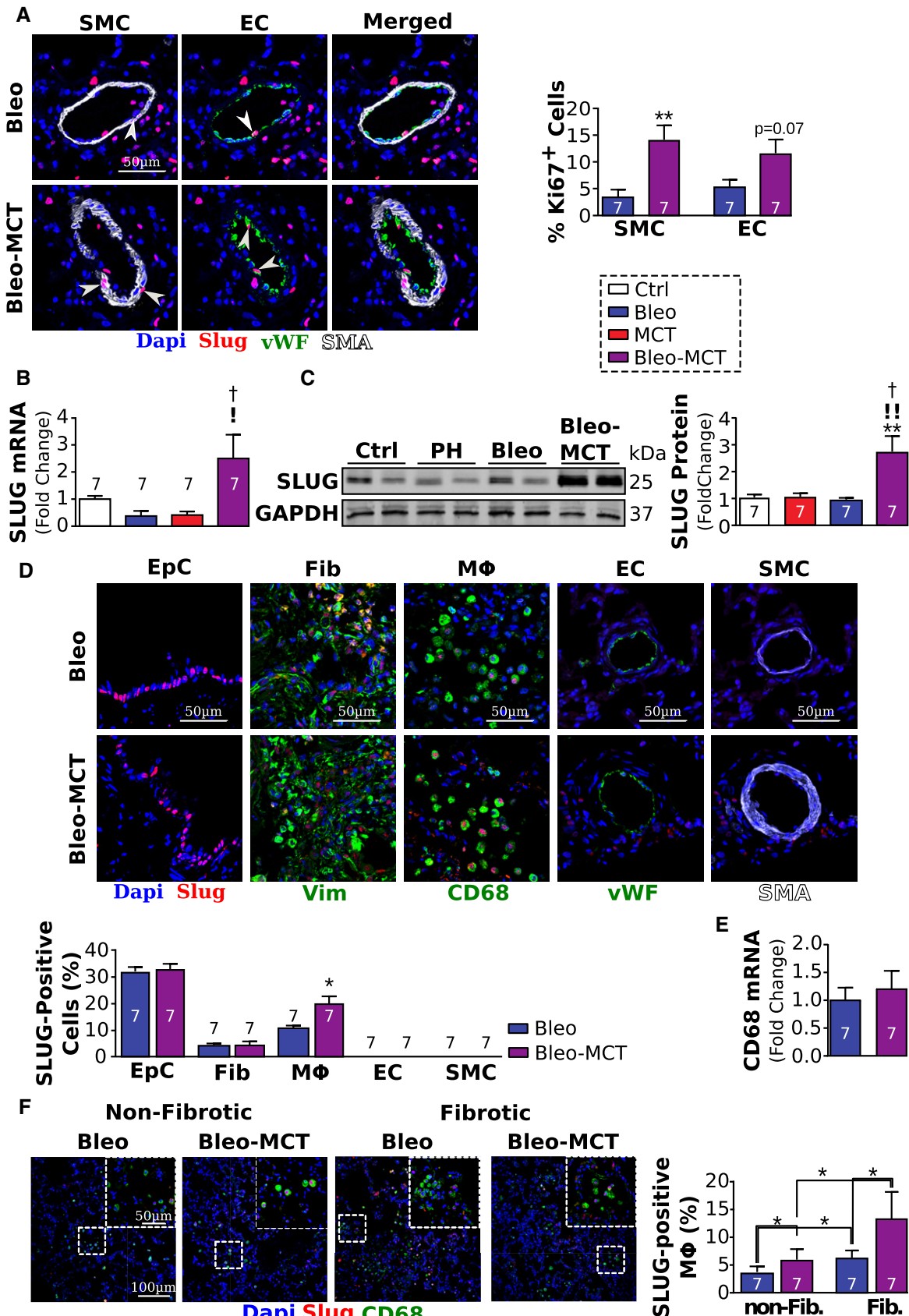

**Figure 5.**

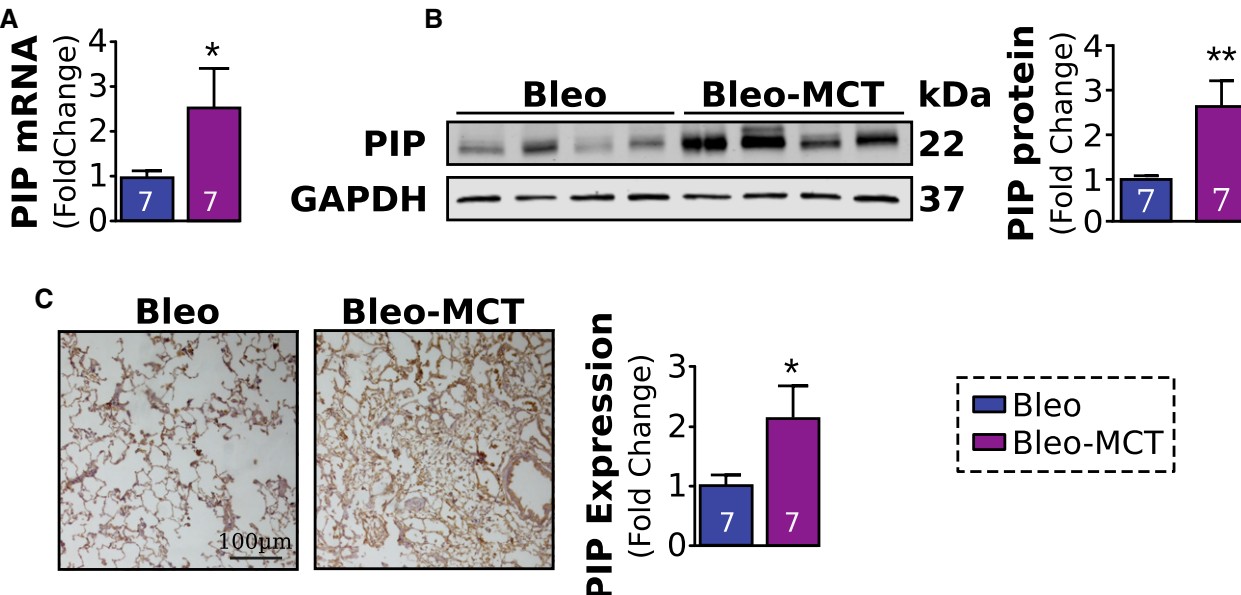

**Figure 6. Increased PIP expression in the lungs of PF-PH compared to PF rats.**

A, B   Relative expression of PIP mRNA (A) and protein (B) normalized to GAPDH.

C      Representative images and PIP quantification.

Data information: Values are expressed as mean ± SEM. The number of samples per group for each experiment is included within each bar graph (t-test; *P < 0.05, **P < 0.01).

Si-Slug rats compared to Si-Scrm rats in both fibrotic and non-fibrotic areas (Fig 7J and K). This decreased vascular wall thickness was concomitant with a significant reduction in Ki67-positive nuclei from EC and SMC (Fig 7L).

Furthermore, Si-Slug treatment significantly reduced the expression of Slug expression by the macrophages but had no effect on CD68 expression (Fig 8A and B). In fact, Slug inhibition resulted in downregulation of Slug in both fibrotic and non-fibrotic areas (Fig 8C). Consistent with this change in Slug expression, levels of PIP mRNA and protein were significantly decreased in Si-Slug compared to Si-Scrm (Fig 8D–F).

In summary, Slug inhibition in the presence of a PH stimulus decreases PH severity in rats with pre-existing PF by decreasing vascular remodeling and PIP-mediated vascular wall cell proliferation.

## Discussion

In this study, we report a distinct pulmonary vascular remodeling pattern that differentiates PF-PH and PF patients. In PF-PH patients, we observed increased vascular wall thickening in both non-fibrotic and fibrotic areas, whereas PF patients mainly exhibit vascular wall thickening in fibrotic areas. We also identified the transcription factor, Slug, which is significantly upregulated in the lung macrophages of PF-PH compared to PF patients. In addition, we report that Slug upregulation in the lungs of PF-PH patients is associated with increased vascular cell proliferation via its transcriptional target PIP. Our study and others demonstrate that the well-accepted experimental model of PF induced by bleomycin alone does not recapitulate histological characteristics of end-stage

PF-PH patients (Jarman et al, 2014). Thus, we developed a combined model of PF-PH in rats that mimics vascular remodeling (in both fibrotic and non-fibrotic areas of lung) and expression of Slug and PIP in end-stage PF-PH patients. Our data show that while Slug is expressed in epithelial cells, macrophages, and fibroblasts in the lung, Slug expression is only significantly higher in PF-PH compared to PF in macrophages both in rat lungs similar to humans. Using this model, we found that Slug inhibition in the presence of a PH stimulus decreases PH severity in rats with pre-existing PF by decreasing vascular remodeling and PIP-mediated vascular wall cell proliferation.

In the present study, we showed a correlation between fibrosis severity and vascular wall thickness in end-stage PF-PH patients (Fig 1F and G). This correlation was previously described in PF patients (Parra et al, 2005; Mlika et al, 2015) and agrees with the data presented here. The presence of vascular remodeling in the non-fibrotic areas of the lung in PF-PH patients was suggested by others, yet never previously quantified (Heath et al, 1968; Kim et al, 2010). Colombat et al, 2007 recently found a significant positive correlation between the macroscopic extent of lung fibrosis and mean pulmonary artery pressure in idiopathic PF patients, suggesting that the increased pressure is only driven by lung fibrosis. However, several other studies did not find any significant differences between lung fibrosis in PF-PH patients compared to PF (Nadrous et al, 2005; Zisman et al, 2007; Nathan et al, 2019). To examine whether the increased vascular remodeling in PF-PH is due to an increase in lung fibrosis, we compared the Ashcroft score as well as the expression of pro-fibrotic markers between PF-PH and PF patients. We did not find any significant differences between Ashcroft score and the expression of fibronectin, collagen I, and collagen III between PF and PF-PH patients (Fig 1A–D) suggesting

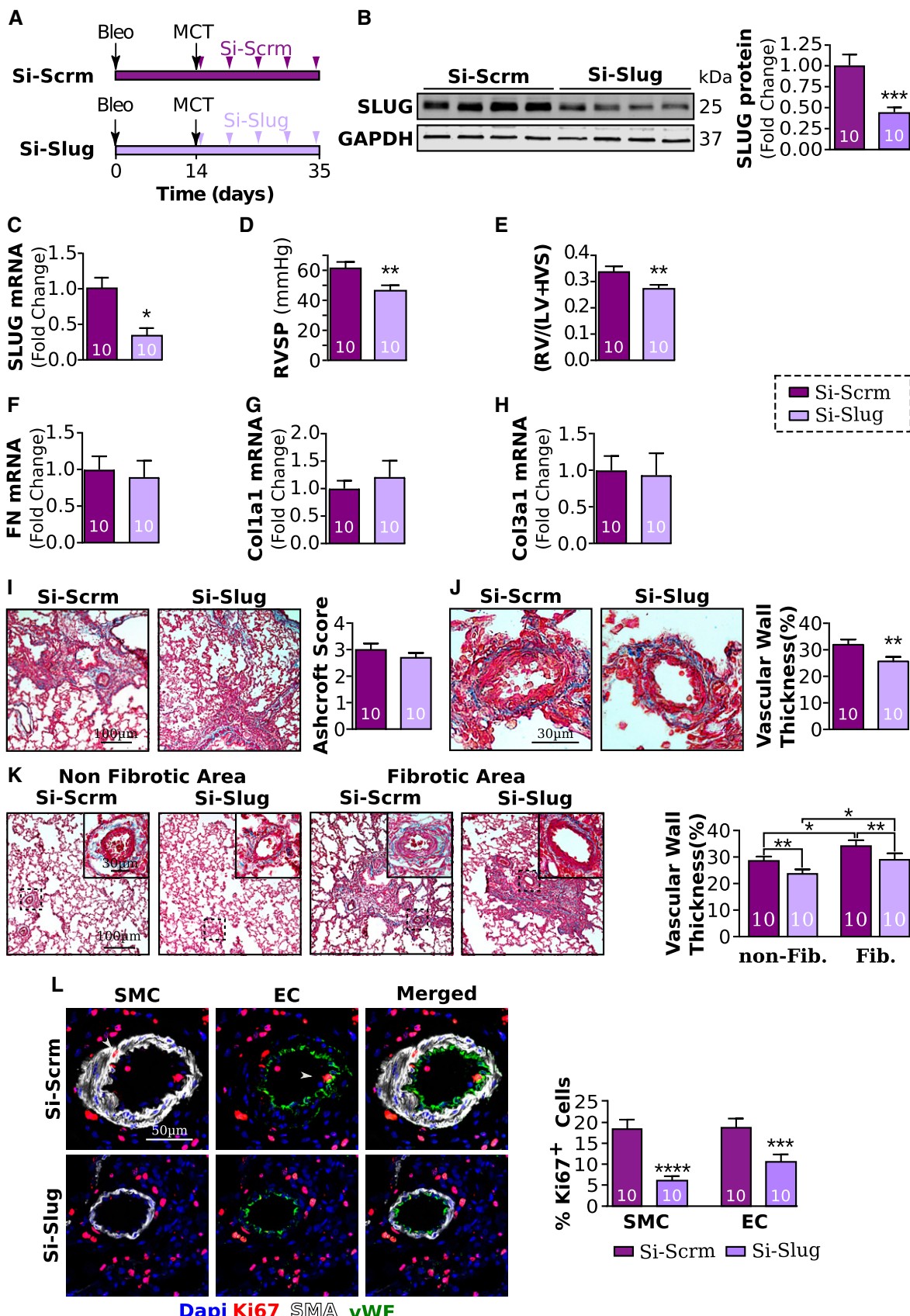

**Figure 7.**

◀

that the increase in vascular remodeling in PF-PH is driven by activation of specific molecular pathways. In fact, other groups demonstrated distinct molecular differences between PF and PF-PH patients suggesting the activation of specific molecular pathways in driving PH (Harari *et al*, 1997; Leuchte *et al*, 2004; Nadrous *et al*, 2005; Mura *et al*, 2012). For example, Mura *et al*, 2012 demonstrated a heightened inflammatory response and increased cell proliferation were implicated in this process.

In the context of left heart disease, it has been shown that about 50–80% of patients develop PH (group 2) due to a passive increase of pulmonary vascular pressure. Overtime, in a subset of these patients, PH will be sustained via an active mechanism leading to vascular remodeling (Guazzi, 2014; Breitling *et al*, 2015). We believe a similar paradigm could apply to PF-PH patients, where PF patients may initially show increased pressure arising from lung fibrosis that, over time, may induce specific molecular changes in the vascular walls. The cause of these molecular and cellular changes remains largely unknown, although various environmental and genetic factors may play a role and should be investigated further in future studies.

Bleomycin has been used for decades to induce experimental model of PF. Mice on chronic low dose of bleomycin have also been shown to develop some degree of PH (Collum *et al*, 2017; Mertens *et al*, 2018). The development of PH in mice on chronic dose of bleomycin was mitigated by hyaluronan synthase inhibition (Collum *et al*, 2017) and also in smooth muscle cell adenosine A2b receptor knockout mice (Mertens *et al*, 2018). Since mice never develop severe pulmonary vascular remodeling as in rats and do not mimic histopathological features of human PH, we developed our combined model of PF-PH in rats. We found that rats treated with bleomycin alone do not recapitulate the histological and molecular features of PF-PH patients. For example, bleomycin alone only induces vascular remodeling within the fibrotic areas, and Slug is not upregulated in the lungs of rats in bleomycin alone group (Fig 5C; Jayachandran *et al*, 2009). In addition, lung expression levels of Slug and PIP in rats were only upregulated in combined PF-PH compared to PF alone as in human lungs. Therefore, we developed a new animal model, by sequential administration of bleomycin and MCT in rats, that better recapitulates the histological differences (vascular remodeling in both fibrotic and non-fibrotic areas) and molecular differences (Slug and PIP upregulation) observed in our PF-PH patients (Figs 4 and 5). Our novel animal model provides a new tool to understand group 3 PH pathology.

Modification of the vascular ECM is part of the vascular remodeling mechanism seen in PH and has recently been investigated in PF-PH patients (Milara *et al*, 2016). However, in our study we did not find any significant differences between collagen I, collagen III, and fibronectin expression in PF-PH compared to PF in both rats and humans. The lack of changes in fibrotic markers in the whole lung tissue does not indicate the absence of increased deposition of vascular ECM. These proteins are also involved in lung fibrosis so local vascular changes of the ECM may be masked by the high concentration of these proteins in the lung parenchyma.

Transcription factor Slug has been implicated in PAH, as well as in COPD (Ranchoux *et al*, 2015; Coll-Bonfill *et al*, 2016; Hopper *et al*, 2016). In our study, we are investigating the role of Slug for the first time in PH secondary to PF, for which the clinical presentation as well as the underlying mechanism of PH is different from PAH and COPD patients. We found expression of Slug is significantly upregulated in the lungs of PF-PH compared to PF patients. In the lung, Slug is highly expressed in the epithelial cells and to lesser extent in macrophages, fibroblasts, and EC (Figs 2D and 5D). While previous studies show Slug expression in SMC in PAH patients (Coll-Bonfill *et al*, 2016), we did not detect Slug expression in SMC in PF-PH patients nor in our animal model. Taken together, these data strengthen the molecular differences between PF-PH and PAH pathologies. While Slug expression was found in multiple cell types, our data show Slug upregulation in PF-PH compared to PF is driven by its significantly higher expression in macrophages in both human and rat lungs. Although the inflammatory nature of MCT in the combined PF-PH rats could be a confounding factor compared to PF rats who receive bleomycin alone, there were no significant differences in CD68 transcript levels between PF and PF-PH rats suggesting that the sequential administration of bleomycin and MCT does not induce an increased recruitment of macrophages in our study. The expression of Slug in inflammatory cells has already been shown in other diseases (Inoue *et al*, 2002; Moldovan & Kusewitt, 2007), but, to our knowledge it has never been demonstrated in any form of PH.

To further examine how Slug upregulation may participate in vascular remodeling in PF-PH patients, we reanalyzed publicly available microarray data from PF and PF-PH patients. We used online microarray data as a discovery tool to identify Slug targets that are implicated in vascular remodeling and secreted in the extracellular space. Our cohort and cohort used in online microarray have similar clinical characteristics such as age, sex, and

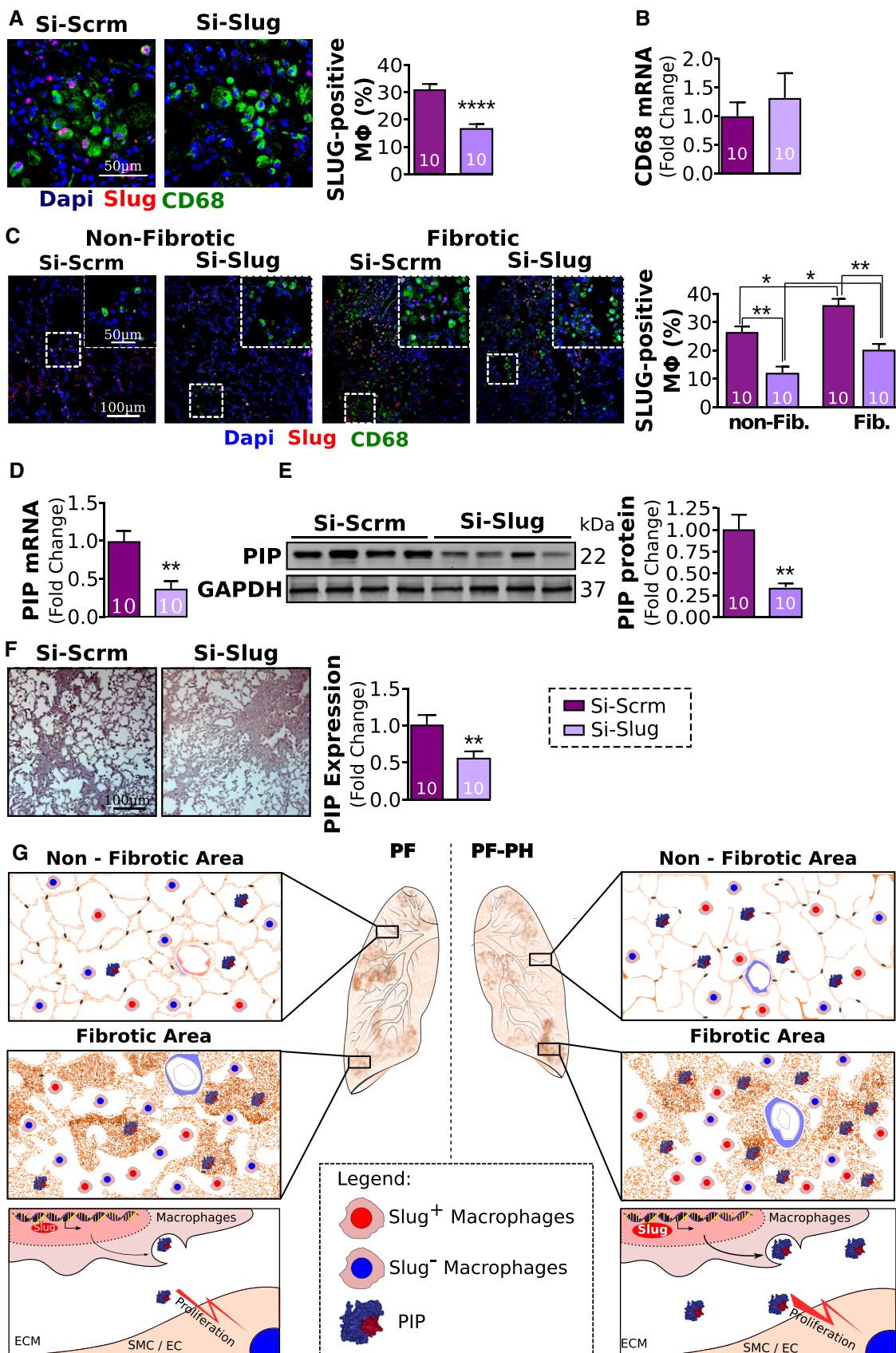

**Figure 8.**

◀

**Figure 8. Slug inhibition reduces vascular remodeling and proliferation of vascular wall cells.**

A    Representative images and quantification of Slug-positive macrophages (Slug, red; CD68 green).
B    Relative expression of CD68 mRNA normalized to GAPDH.
C    Representative images and quantification of Slug-positive macrophages between non-fibrotic and fibrotic areas of the lung.
D, E    Relative expression of PIP mRNA (D) and protein (E) normalized to GAPDH.
F    Representative images and quantification of PIP immunohistochemistry.
G    Proposed mechanism. In pulmonary fibrosis, vascular wall thickening is mainly seen in fibrotic areas of the lung, while in PH secondary to PF, vascular remodeling is seen in both fibrotic and non-fibrotic areas of the lung. The extension of vascular wall thickening to the non-fibrotic areas of the lung in PF-PH is concomitant with an increased expression of Slug and PIP in the lungs in both fibrotic and non-fibrotic areas. At the cellular level, Slug upregulation in macrophages in PF-PH leads to increased PIP expression in the extracellular space, which in turn triggers SMC proliferation leading to vascular wall thickening observed in fibrotic and non-fibrotic areas of the lung (ECM: extracellular matrix; SMC/EC: smooth muscle cells and endothelial cells).

Data information: Values are expressed as mean ± SEM. The number of samples per group for each experiment is included within each bar graph. Statistical tests: panels (A, B, and D–F): t-test; panel (C): ANOVA (*$P < 0.05$, **$P < 0.01$, ****$P < 0.0001$).

forced vital capacity (Mura *et al*, 2012). Our analysis revealed upregulation of 59 transcriptional targets of Slug in PF-PH versus PF patients (Fig 3A–C). Among these genes, we identified PIP, an extracellular matrix protein known to induce cancer cell proliferation (Naderi & Meyer, 2012; Baniwal *et al*, 2013), and confirmed its upregulation in our PF-PH patients (Fig 3D–F). The fact that we were able to validate PIP upregulation in our own cohort further strengthens the implication of PIP in PF-PH patients. Furthermore, our data demonstrate the role of PIP in vascular remodeling by promoting SMC and EC proliferation isolated from pulmonary artery *in vitro* (Fig 3G). Our data show PIP does not promote fibroblast proliferation, and therefore, PIP may not directly participate in the development of pulmonary fibrosis (Fig 3G). Furthermore, reduced expression of PIP by Slug-siRNA did not have any effect on the Ashcroft score or the expression of the fibronectin, collagen I, and collagen III, further supporting the absence of a role for Slug/PIP axis in fibroblast remodeling (Fig 6E–H). Our data support the presence of histological and molecular differences

**Table 1. Clinical characteristics of the patients.**

|  | Ctrl (*n* = 4) | PF (*n* = 14) | PAH (*n* = 7) | PF-PH (*n* = 14) |
|---|---|---|---|---|
| Age (years) | 25 ± 4.5 | 61 ± 10 | 43 ± 11 | 62 ± 9 |
| Sex Male, *n* (%) | 4 (100) | 9 (64) | 0 (0) | 8 (57) |
| Diagnostic, *n* (%) |  | IPF, 7 (50) | IPAH, 7 (100) | IPF-PH, 8 (57) |
|  |  | ILD, 7 (50) |  | ILD, 6 (43) |
| mPAP (mmHg, *n*) | – | 20 ± 4 (14) | 63 ± 8 (7) | 32 ± 2 (14) |
| CO (l/min, *n*) | – | 5.4 ± 1.2 (13) | 5.5 ± 1.9 (7) | 4.6 ± 1.0 (13) |
| PVR (wood unit, *n*) | – | 2.6 ± 0.7 (9) | 9.10 ± 5.9 (3) | 4.8 ± 1.7 (9) |
| Time lapse (month) | – | 7.8 ± 5 (14) | 13.5 ± 7.5(6) | 11 ± 10 (14) |
| 6MWD (m, *n*) | – | 221 ± 123 (11) | 230 ± 105 (6) | 194 ± 129 (11) |
| FVC (L, *n*) | – | 2.13 ± 1 (13) | – | 1.87 ± 1 (14) |
| FVC (%predicted, *n*) | – | 48.9 ± 19 (13) | – | 49.9 ± 19 (14) |
| FEV1(L, *n*) | – | 1.78 ± 0.8 (13) | – | 1.57 ± 0.7 (14) |
| FEV1 (% predicted, *n*) | – | 54.1 ± 22 (13) | – | 56.0 ± 19 (14) |
| FEV1/FVC | – | 83.8 ± 5.7 (13) | – | 85.6 ± 8.4 (14) |
| DLCO (ml/min/mmHg$^2$, *n*) | – | 6.4 ± 2.4 (11) | – | 6.6 ± 2.9 (12) |
| DLCO (% predicted, *n*) | – | 24.4 ± 1.8.1 (12) | – | 26.5 ± 9.4 (11) |
| Oxygen saturation (%, *n*) | – | 96.1 ± 3.1 (14) | – | 97.1 ± 2.5 (14) |
| Oxygen use (L) | – | 3.53 ± 1.9 (13) | – | 4.9 ± 1.9 (14) |
| Medication, *n* (%) |  |  |  |  |
| Antifibrotic | – | 1 (7)) | 0 (0) | 2 (14) |
| Prostacyclin | – | 0 (0) | 7 (100) | 1 (7) |
| ET1R inhibitor | – | 1 (7) | 6 (86) | 1 (7) |
| PDE5 inhibitor | – | 0 (0) | 7 (100) | 4 (28) |

Definition of abbreviations: mPAP: mean pulmonary arterial pressure; CO: cardiac output; PVR: pulmonary vascular resistance; Time lapse: time between the right heart catheterization and transplant/death. 6MWD: 6-min-walking distance; ILD: interstitial lung disease (other than IPF); IPAH: idiopathic PAH; IPF: idiopathic pulmonary fibrosis; PAH: pulmonary arterial hypertension; PH: pulmonary hypertension; PF: pulmonary fibrosis. Values are expressed as mean ± SD.

between PF-PH and PF patients and give the first evidence of the role of Slug in macrophages in promoting vascular wall cell proliferation through the presence of its target PIP in the extracellular space in PF-PH lung.

While the molecular mechanism by which PIP promotes SMC and EC proliferation in the setting of PF-PH is not known, PIP has been shown to induce cell proliferation by multiple pathways in cancer. Recently, it was demonstrated that PIP is an aspartyl proteinase able to cleave fibronectin (Caputo et al, 2000). In turn, fibronectin fragments induce cell proliferation (Naderi & Meyer, 2012). In another study, Baniwal et al (2013) demonstrated that PIP inhibition decreases cell proliferation and migration through the focal adhesion kinase, which is known to be increased in the remodeled vessels of PF patients (Lagares et al, 2012), and promotes pulmonary artery SMC migration and proliferation in PAH (Paulin et al, 2014). These evidence highlight the role of PIP in cell proliferation through various mechanisms, all of which could potentially be relevant in PF-PH pathology and should be investigated in future studies. Although we demonstrated the role of PIP in SMC and EC proliferation, we also identified six other transcriptional targets of Slug found to be upregulated in PF-PH versus PF patients that are involved in proliferation and are secreted into the extracellular space. These genes could also drive mechanistic differences between PF-PH and PF patients. For example, we identified the latent-transforming growth factor beta-binding protein 4 (LTBP4) which binds and activates TGF-β in the extracellular matrix (Koli et al, 2004). This pathway may play an important role in PF-PH development since TGF-β is implicated in lung fibrosis (Tatler & Jenkins, 2012) and PH (Botney et al, 1994; Gore et al, 2014). The insulin growth factor binding protein 2 (IGFBP2), another protein upregulated in PF-PH, is known to control the activation of insulin growth factors and is also implicated in PH pathogenesis (Guiot et al, 2016). This protein is also a predictive marker of idiopathic PF severity (Guiot et al, 2015, 2016) which highlights its potential role in PF-PH pathology.

In agreement with our in vitro experiments showing exogenous PIP is able to induce proliferation in both SMC and EC, our in vivo experiments also show Slug inhibition is associated with decreased expression of PIP and decreased vascular remodeling leading to a decreased PH severity (Figs 7J and 8D and E). These experiments together support the role of Slug/PIP axis in promoting pulmonary vascular remodeling and demonstrate the therapeutic potential of Slug inhibition in a pre-clinical model of PF-PH (Figs 6 and 7).

Our research identifies histological and molecular differences between PF-PH and PF patients and gives the first evidence of the implication of Slug/PIP axis in PF-PH pathophysiology. Whether our findings can be applied to other forms of PF remains to be seen. As in all animal models, our combined animal model of PF-PH has some limitations. While we found this model represents the aspects of end-stage disease in humans studied here, further investigation is necessary to identify whether it represents other histological and molecular characteristics of PF-PH as well.

This study highlights histological and molecular differences between end-stage PF and PF-PH patients by the presence of vascular remodeling in non-fibrotic areas of the lung in end-stage PF-PH patients and implicates the Slug/PIP axis in vascular remodeling (Fig 8G). Furthermore, we demonstrate that Slug inhibition

reduces PH severity in a pre-clinical model of PF-PH, which may pave the way toward a better understanding of this debilitating disease.

# Materials and Methods

### Human tissue samples

Prior to collecting patient lung samples, we obtained institutional review board approval from the Office of the Human Research Protection Program (OHRPP) at UCLA and informed consent from all subjects and confirmed that the intended experiments conformed to the principles set out in the WMA Declaration of Helsinki and the Department of Health and Human Services Belmont Report. Lung sections were obtained from explanted lungs at end stages of lung diseases (at the time of lung transplant or deceased) including pulmonary fibrosis (PF, $n = 14$), pulmonary arterial hypertension (PAH, $n = 7$), pulmonary hypertension secondary to pulmonary fibrosis (PF-PH, $n = 14$), and control (with no history of lung disease, Ctrl, $n = 4$). Out of the 14 PF and PF-PH patients, 3 had no non-fibrotic area and therefore were excluded for comparison between non-fibrotic and fibrotic areas. For Ctrl and PAH patients, we only had access to slides. For PF and PF-PH patients, out of 14 patients in each group, we had access to the slides and lung samples from 6 patients in each group and we only had access to slides for remaining 8. Clinical characteristics of patients and control subjects are given in Table 1. In addition, Appendix Table S1 provides individual values of the parameters listed in Table 1 for each patient. Considering mPAP > 40 mmHg is one of the most accepted criteria to define severe PH in PF patients (although there is still no consensus to define mild and severe PH, Trammell et al, 2015), 12 of our PF-PH patients (85%) would be classified as mild PH, and two patients with mPAP of 44 and 47 mmHg would be classified as severe PH. We included these two patients in our study since all of their other parameters did not significantly differ from the other PF-PH patients in this group (Table 1 and Appendix Table S1).

### Animal model

Male, Wistar rats (~200–250 g) were purchased from Charles River Laboratories. All experimental protocols received Animal Research Committee (ARC) at UCLA institutional review and committee approval. Animals were divided into four groups ($n = 7$/group): control (Ctrl), pulmonary fibrosis (PF), pulmonary hypertension (PH), and PF secondary to PH (PF-PH). At day 0, rats in the PF and PF-PH groups received an intratracheal instillation of bleomycin (2.5 mg/kg) (Tomos et al, 2017), whereas the Ctrl and PH groups received a PBS instillation. At day 14, the PH and PF-PH groups received a subcutaneous injection of monocrotaline (MCT, 60 mg/kg), whereas the Ctrl and PF groups received a subcutaneous injection of PBS (Fig 4E).

To examine whether Slug inhibition can decrease PH severity in rats with pre-existing PF, rats were randomly divided into the control and Slug inhibition groups (10 rats/group). All rats received bleomycin at day 0 and MCT at day 14. Starting at day

17, rats received five intratracheal nebulizations (twice a week for 3 weeks) of either a small interfering RNA (siRNA) directed against Slug (Si-Slug group; Dharmacon A-093398-13-0050; 5 nmol) or scramble siRNA (Si-Scrm; Dharmacon D-001910-01-50; 5 nmol) (Fig 7A).

### Tissue preparation, staining, and imaging

Lungs were fixed in 4% paraformaldehyde, immersed in 20% sucrose, mounted in OCT compound, and sectioned with microtome at 5 μm. Paraffin-embedded control lung sections, obtained from the UCLA pathology laboratory, were sectioned at 5 μm. Images were acquired using a confocal microscope (Nikon).

After removing OCT or paraffin, all slides were boiled for 30 min in citrate buffer as an antigen retrieval step. For nuclear staining, slides were further immersed in cold methanol for 5 min to ensure nuclear permeabilization. Slides were then blocked with PBS 1× with 0.2% Triton and 10% goat serum for 1 h. Primary antibody was applied to the slides overnight at 4C in PBS 1× with 0.2% Triton and 5% goat serum. Secondary antibody was incubated for 2 h at 37C. All co-immunofluorescence was done in sequence. Primary antibodies used were as follows: Ki67 (Millipore, AB9260, 1:200); Slug in rats (Abcam, 27568, 1:500); Slug in human (LS Bio, LS-C175161, 1:200); Cd68 in human (Bio-Rad, MCA5709A647, 1:500); Cd68 in rats (Bio-Rad, MCA341A647, 1:500); aSMA (Novus, NBP2-34522AF647, 1:500); vWF (Abcam, Ab6994, 1:200); vimentin (Cell Signaling, CS3932, 1:200); and PIP (LS Bio, LS-B15418, 1:250). Secondary antibodies used were as follows: biotinylated anti-rabbit (Abcam, ab97062, 1:500); streptavidin-HRP (Abcam, ab7403, 1:1,000); anti-mouse (594) (Invitrogen, A11005, 1:1,000); anti-mouse (488) (Invitrogen, A11001, 1:1,000); anti-rabbit (594) (Invitrogen, A11012, 1:1,000); and anti-rabbit (488) (Invitrogen, A11008, 1:1,000).

### Western blot analysis

Proteins were extracted using RIPA buffer (Sigma, R0278) supplemented with protease inhibitor (cOmplete Mini, 11836153001) and phosphatase inhibitor (Roche, 04906845001). Protein concentration was measured using a Bradford assay (Sigma, B6916). 20 μg of protein was loaded per well of the gel. Primary antibody was applied to the membrane overnight at 4C, and secondary antibody was applied at room temperature for 1 h. The membrane was revealed using the Odyssey apparatus from Li-Cor. Western blot quantification was performed using Image Studio software. The intensity of each band was assessed for each sample and antibody. GAPDH intensity was used to normalize the protein loading. Primary antibodies used were as follows: Slug for rats (Abcam 27568, 1:1,000); Slug for human (LS Bio, LS-C175161, 1:1,000); GAPDH (Novus, NB300-327, 1:2,000); and PIP (LS Bio, LS-B15418, 1:1,000). Secondary antibodies used were as follows: anti-mouse (800) (Li-Cor, 926-68070, 1:5,000) and anti-rabbit (680) (Li-Cor, 926-68070, 1:5,000).

### Real-time PCR

Total RNA from lungs was isolated with TRIzol and reverse-transcribed with poly-dT primers using Omniscript reverse transcription kit (Qiagen). Real-time PCR was performed on polyA+ cDNA with specific primers using iTaq Universal SYBR® (Bio-Rad). GAPDH was used as a housekeeping gene to normalize the transcript expression of each gene to GAPDH.

Specific primers used in human:
GAPDH (NM_002046) forward: ATCTTCTTTTGCGTCGCCAG, reverse: GGCGCCCAATACGACCAAA; SLUG (NM_003068) forward: CGAACTGGACACACATACAGTGAT, reverse: GAGAGGCCATTGGG TAGCTG; PIP (NM_002652) forward: TCCCAAGTCAGTACGTCCA AA, reverse: CACCTTGTAGAGGGATGCTGC; collagen 1a1 (NM_ 000088) forward: TCTCCCCAGAAGACACAGGAA, reverse: GACT CTCCTCCGAACCCAGT; collagen 3a1 (NM_000090) forward: CTTCTCTCCAGCCGAGCTTC, reverse: TAGTCTCACAGCCTTGCGTG; fibronectin (NM_212482) forward: GTGCCATTTGCTCCTGCAC, reverse: CTCGGGAATCTTCTCTGTCAGC; CD68 (NM_001251) forward: GTCTACCTGAGCTACATGGCG, reverse: GATGAGAGGC AGCAAGATGGA.

Specific primers used in rats:
GAPDH (NM_017008) forward: GTGCCAGCCTCGTCTCATAG, reverse: GGTAACCAGGCGTCCGATAC; SLUG (NM_013035) forward: AGACTCCAGCCCAAGCTTTC, reverse: GCTTTTCCCCAGTGTGTGTTC; PIP (NM_022708) forward: ATACACCGGTTGATGGTGGC, reverse: TGCGACGTTATTAGGGCAGA; collagen 1a1 (NM_053304) forward: GTACATCAGCCCAAACCCCA, reverse: ACAAGCGTGCTGTAGGTGAA; collagen 3a1 (NM_032085) forward: AGCGGAGAATACTGGGTTGA, reverse: TAGCTGAACTGAAAGCCACCA; fibronectin (NM_019143) forward: GGGCTCAATCCAAATGCCTCT, reverse: GCTGGTTCAGGC CTTCGTT; Cd68 (NM_001031638) forward: ACCCGGAGACGACAA TCAAC, reverse: CTTGGTGGCCTACAGAGTGG.

### Ashcroft score quantification

Ashcroft scores were assessed using the modified Ashcroft scale defined by Hubner et al (Hübner et al, 2008) on Masson's trichrome-stained lung sections analyzing at least five fields per slides (10×). Non-fibrotic areas were defined by an Ashcroft score below or equal to two, and fibrotic areas were defined by an Ashcroft score above two.

### Quantification of vascular wall thickness

Vascular wall thickness was assessed on distal pulmonary vasculature (< 100 μm) using at least five arteries randomly selected from each group. For fibrotic and non-fibrotic areas, vascular wall thickness was quantified on at least five arteries randomly chosen from each area. Vascular wall thickness was calculated by averaging the external and internal diameter of transversally cut vessels and determining the mean distance between the lamina elastica externa and lumen in two perpendicular directions as already described (Ruffenach et al, 2016; Rol et al, 2017).

### Quantification of vascular wall cell proliferation

Vascular cell proliferation was assessed on distal pulmonary vasculature (< 100 μm) using at least 5–10 arteries randomly selected from each animal/subject per group. For each quantified vessel, the total number of endothelial cells (vWF-positive cells),

smooth muscle cells (aSMA-positive cells), and double-positive cells (Ki67/vWF and Ki67/aSMA) was quantified to obtain the percent of Ki67-positive endothelial and smooth muscle cells.

### Quantification of PIP staining

PIP staining was analyzed using the ImageJ plugin IHC profiler. All quantification was performed on 5–10 randomly chosen images per slide. Immunohistochemistry was used over immunofluorescence for quantification considering the autofluorescence and high background most often found with fibrotic lung tissue.

### Quantification of SLUG-positive cells

All quantification was performed on 5–10 randomly chosen images per slide. For each cell type, the total number of cells per image and the number of SLUG-positive cells were quantified to calculate the percent SLUG-positive cells.

### Cell proliferation experiments

Primary cultured pulmonary arterial smooth muscle cells, endothelial cells, and fibroblasts from healthy subjects were seeded in a 96-well plate (1,000 cells/well). Cells were treated with 25 or 50 ng of PIP (MBS146246) or the vehicle. Three days later, the CC8 assay was performed to assess cell proliferation following the manufacturer's protocol (Dojindo Laboratories, CK04). Experiments were performed at least three times.

### Hemodynamics and gross histological evaluation

Right ventricular systolic pressure (RVSP) was measured in anesthetized animals by directly inserting a catheter into the right ventricle (RV). The heart was then removed, and the RV wall, left ventricular (LV) wall, and interventricular septum (IVS) were dissected, and the ratio of RV to LV plus IVS weight [RV/(LV+IVS)] was calculated as an index of RV hypertrophy.

### Microarray analysis

Publicly available human microarray data (Mura *et al*, 2012) were analyzed using the Limma package (Ritchie *et al*, 2015) to define a list of extracellular Slug transcriptional targets known to promote cell proliferation. The list of differentially expressed genes was cross-referenced with a list of Slug, known or predicted, targets from the Harmonizome [6] to define the list of Slug-controlled genes differentially expressed in PF-PH (Appendix Table S2). This list was sorted based on gene ontology and published literature to define genes that are known to be implicated in cell proliferation (cell growth GO:0030307; cell proliferation GO:0008283; regulation of cell proliferation GO:0042127) (Appendix Table S3) and to be present in the extracellular space (extracellular region GO:0005576; extracellular space GO:0005615; extracellular matrix GO:0031012) (Appendix Table S4). Finally, the two lists were cross-referenced to identify genes that are transcriptionally controlled by Slug, implicated in cell proliferation, and present in the extracellular space.

### The paper explained

#### Problem

Pulmonary hypertension (PH) is a major complication of pulmonary fibrosis (PF) but there is no approved therapy. The molecular mechanism of why pulmonary hypertension (PH) develops in this patient population remains understudied and poorly understood.

#### Results

In this study, we show that vessel walls are thickened in both fibrotic and non-fibrotic areas of the lungs of patients with PH secondary to PF but this thickening is mainly restricted to the scarred areas in patients with PF alone. In pulmonary hypertension secondary to pulmonary fibrosis (PF-PH), expression of transcription factor Slug is increased only in macrophages, leading to increased expression of its transcriptional target prolactin-induced protein (PIP). PIP is an extracellular matrix protein that induces proliferation of pulmonary arterial smooth muscle cells. In a pre-clinical model of PF-PH, Slug inhibition reduced the severity of PH.

#### Impact

Our study provides mechanistic insights into the pathological changes seen in PH as a complication of PF. We highlight the transcription factor Slug as a potential novel therapeutic target for this devastating disease which currently has no approved treatment.

### Statistics

For comparison between two groups, we used an unpaired *t*-test. We used an ANOVA test to compare more than two groups/conditions. One-way ANOVA was used for comparing Ctrl, PF, PAH, and PF-PH patients/rats, and two-way ANOVA was used when comparing fibrotic and non-fibrotic areas of the lung from PF and PF-PH patients/rats, followed by a Sidak multiple comparisons test to compare a set of means. The normality of the data was verified using the Shapiro–Wilk test, and the Brown–Forsythe test (modified Levene's test) was used to verify homogeneity of variances. When homogeneity was not fulfilled, values were log-transformed to stabilize variances and statistical analyses were performed on these log transformations. Values below 5% ($P < 0.05$) were considered statistically significant. The *P*-values are listed in Appendix Table S5. All analyses were performed using GraphPad (Software, San Diego, CA, USA).

**Expanded View** for this article is available online.

### Acknowledgements

This study is supported by the National Institutes of Health (R01HL129051 ME and STR), the American Heart Association (17POST33670424 RG, 17PRE33420159 CMC), and the Foundation for Anesthesia Education and Research (SU).

### Author contributions

GR, SU, MV, NC, JH, SS, SM, CMC, AA, STR, RS, GF, and ME were responsible for collecting, analyzing, and interpreting the data; GR, MV, and ME drafting the work; GR, MV, JH, CMC, STR, RS, GF, and ME revising it critically for important intellectual content; GR, SU, MV, NC, JH, SS, SM, CMC, AA, STR, RS, GF, and ME final approval of the version to be published. ME supervised the study.

## Conflict of interest

The authors declare that they have no conflict of interest.

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
