## [Review Process File · EMBO Molecular Medicine]

Histological Hallmarks and Role of Slug/PIP Axis in Pulmonary Hypertension Secondary to Pulmonary Fibrosis

G. Ruffenach, S. Umar, M. Vaillancourt, J. Hong, N. Cao, S. Sarji, S. Moazeni, C.M. Cunningham, A. Ardehali, S.T. Reddy, R. Sagar, G. Fishbein, M. Eghbali

Review timeline:

Submission date:	12 November 2018
Editorial Decision:	19 December 2018
Authors' Appeal:	14 January 2019
Editor's Response:	15 January 2019
Revision received:	25 May 2019
Editorial Decision:	30 June 2019
Revision received:	31 July 2019
Accepted:	1 August 2019

Editor: Lise Roth

Transaction Report:

1st Editorial Decision

19 December 2018

As you will see from the enclosed reports, the referees acknowledge the potential interest of the topic. However, they remain unsatisfied and are not convinced that as it stands, the main conclusions of the study are well supported by the data, particularly regarding the expression of Slug in different cell types. Considering the substantial points raised and the overall rather low level of support provided by the reviewers, I am afraid I see little choice but to return the manuscript to you at this point with the decision that we cannot offer to publish it.

I am sorry that I could not bring better news this time and hope that the referee comments are helpful in your continued work in this area.

***** Reviewer's comments *****

Referee #1 (Comments on Novelty/Model System for Author):

The use of IHC is flawed-the signals required to analyze the location and action of slug/PIP on vascular remodeling would be better captured using confocal analysis of lung sections. The IHC is very poor and the counterstain is weak, making appreciation of the pathology very hard to the naked eye.

Referee #1 (Remarks for Author):

1. Patient characteristics: There is no data regarding the pulmonary mechanical properties of these patients. How bad was their restrictive disease? Did they had obstructive lung disease as well? How many were on oxygen? We need more details. I strongly suggest providing information on: 1. FVC, FEV1 and FEV1/FVC, 2) DLCO, 3) oxygen saturation, 4) oxygen use.

2. These IPF-PH patients seem to have mild PH, the most common form. Yet, there is a subgroup that has PH out of proportion to the underlying fibrosis. This must be carefully pointed out as this subtype appears to be different (less severe PFTs, very elevated mPAP).
3. Figure 1F and G. The vascular pathology seen in severe fibrosis of PF-PH is hard to compare against its corresponding PF alone since it is located within a fibrotic lesions while the PF is located in a less scarred zone. The authors should ensure that they are comparing areas of similar fibrotic burden before quantifying vascular remodeling. This raises concerns regarding the data output on this section of the study.
4. Figure 2A. The Ki67 label is hard to interpret. It looks as if the signal comes mostly from endothelial cells rather than SMC or fibroblasts. I request that the authors do IF and label bot EC and SMC along with ki67 to do colocalization studies. Also, severity of the remodeling is not comparable to that seen in Figure 1.
5. Figure 2B. Counterstain is very faint and makes appreciation the vascular wall anatomy very hard. Please follow my request for IF.
6. Figures 2D-G: It is very hard to appreciate cell number, location and position in Figures 2D and G to be comfortable concerning the analysis. Where is the inset region located in the lung section. There also seems to be much more macrophages in figure E compared to what you see on D and G. Is this only because of the greater magnification? It is hard for me to really accept the quantification based on these poor quality images.
7. Figure 3. The authors make use of public data to look at genes associated with Slug. A major problem here is that we don't know the clinical characteristics of these patients nor do we know how they compare with their discovery cohort. Is slug something that responds differently in the setting of mild vs. severe restriction? Is it independent of IPF severity? This information is mandatory to establish relevance of their discovery to clinical setting.
8. Figure 3D-F: IHC for PIP appears to be all over the PF lungs (PH or not). Why are we looking only at SMCs? The ECs and fibroblasts are also part of the path-biology of vascular remodeling in this setting as they are vital components of the vessel wall. Please perform proliferation/apoptosis studies using all 3 cell types. It would also add greater impact to your paper if you have PBMCs from IPF patients to look at the Slug/PIP production via FACS and in co-culture assays.
9. Figure 5. Once again, the IHC is very problematic. I zero in on the IF because it allows me to appreciate the slug signal better (macrophages are notorious for nonspecific stains in IHC) but I can't rule out that slug is not being expressed in other parts of the lung. Are we POSITIVE that slug is only being expressed in macrophages?
10. Figure 6 and 7. The Slug siRNA should affect many cells since it is not targeted to macrophages. This leads me to ask again: are we positive that slug is only located to macrophages? The authors make no effort to characterize whether slug expression is down in the lungs using qPCR/WB or to purify macrophages and measure slug/PIP production after treatment. The remodeled vessels do not look very different to my eye which again raises concerns regarding the quantification used. n

Referee #2 (Remarks for Author):

Ruffenach and colleagues show histological differences in vascular remodeling in patients with pulmonary hypertension secondary to pulmonary fibrosis compared with the lungs of patients with pulmonary fibrosis, which is quite interesting considering that one of the main characteristics of pulmonary hypertension is the increase in remodeling of the pulmonary vasculature and the increase of cell proliferation that is also being demonstrated in this article. On the other hand, the authors show an increase in Slug and PIP, two proteins for which their role during PH development had already been demonstrated in previous works.

The most interesting aspect of this manuscript is the proposal of a new model of pulmonary hypertension derived from pulmonary fibrosis and that the authors demonstrate the increase in RVSP and RVH values, as well pulmonary fibrosis markers such as collagen and fibronectin,

resulting interesting for the future analysis of physiological and molecular mechanisms related to pulmonary hypertension secondary to pulmonary fibrosis.

However, the manuscript needs additional set of experiments to compile these interesting findings.

Major comments

Figure 1 B, C and D, Include the expression levels in control and PAH will give more strength to the results if they match with the results in figure 4 I, J and K.

Figure 2 and 3 D: Include Western blot analysis for Slug and PIP in lung homogenate will be more informative.

Figure 4: Panel B,C,D the authors are showing only 4 rats after Bleo treatment vs 7 controls while in the other set (panels F and G) the n number is the same. I would like to know the main reason to use a less sample number and ask for the increase of the n after bleo treatment to have the same number. Since there are post-transcriptional regulation mechanisms, it would be important to show the protein levels of fibronectin and collagen (panel I, J, K)

The authors are not showing enough data to prove the molecular mechanism through Slug via PIP is regulating PASCAM proliferation; since Slug has more targets that could play a role during PF-PH vascular remodeling; I recommend the isolation of PASCAM cells after Bleo, MCT and siSlug with the respective controls and perform BrdU experiment and the measurement of PIP mRNA expression and the protein levels of Slug and PIP via western blot analysis.

Which pathways they suggest are increasing the vascular remodeling in PF-PH?

Vascular remodeling is not only due to the increase in cell proliferation, it is also due to the production of ECM components, however the author didn't find any change in the expression of CollI, CollIII and fibronectin, may be interesting analyze the expression levels of other ECM proteins.

Page 13 line 2 the data are not enough to assert the role of ECM in cellular communication between macrophages and vascular cells in PF-PH, the expression of ECM proteins doesn't change and the results show an increase in wall thickening in non-fibrotic and fibrotic areas.

Figure 8: The authors found an increase in the expression of Slug in macrophages, however; in previous reports was prove that also epithelial cells have a high expression of Slug in PF. Which are the expression levels of Slug in PF-PH vs PF? The expression of Slug in epithelial cells could have the same effect over PIP and proliferation in PASCAM.

Minor comments

Figure 7: (B) and (C) are interchanged in the foot note. D) Quantification of Slug..... But in the plot is write PIP

Referee #3 (Comments on Novelty/Model System for Author):

In this particular study, the emphasis is on SLUG. Others have shown that mice that overexpress an inducible Slug transgene are viable and appear phenotypically normal; although many of these mice die from cardiac hypertrophy as adults (Perez-Mancera PA, et al Cytogenet Genome Res. 2006;114(1):24-29). Also SLUG has been shown in the mouse hypoxia SUGEN model of PH to be associated with the proliferative phenotype of pulmonary arterial smooth muscle cells (PASCAMs) and vascular remodelling (Coll-Bonfill et al PLOS ONE 11(7): e0159460)

Would it have been better to use the inducible SLUG overexpressing mouse in conjunction with bleomycin to investigate and dissect the SLUG/PIP mechanism in PF-PH?

Referee #3 (Remarks for Author):

This is an interesting and important paper which addresses the mechanisms of pulmonary hypertension secondary to pulmonary fibrosis (PF-PH). The molecular mechanisms and cellular interactions that result in PF-PH are complex and notoriously difficult to dissect because they involved both the pulmonary interstitium and the pulmonary vasculature. In addition the crosstalk of many cell types (epithelial, mesenchymal, and vascular) make it harder to dissect the important pathways that contribute to the pathogenesis. Any research in this field is therefore of great interest.

This work has been carried out with great care and is very detailed. The authors have developed an animal model of PF-PH using rats by combining two well established models for PF (Bleomycin-induced) and PH (Monocrotaline-induced). As the authors have pointed out, neither of these reflect fully the human conditions and are therefore flawed. However, they do provide insights to understanding mechanism.

The authors have shown that the transcription factor SLUG (Snai2) is upregulated in both PF-PH patient lungs and in the rat model. They then demonstrated that inhibition of SLUG by administration of siRNA resulted in decreased pulmonary vascular remodelling, vascular pressures but did not alter the extent of pulmonary fibrosis. They went on to provide some evidence and propose a mechanism by which prolactin-induced protein (PIP), a SLUG downstream target, is secreted by macrophages and induces increased smooth muscle cell proliferation in both non-fibrotic and fibrotic areas of the PF-PH lung. In contrast, the levels of PIP secretion in the PF lungs are lower in non-fibrotic areas and therefore there is no accompanying PH.

Comments:

1) Animal model

The rationale of developing this animal model is correct and it is a welcome addition to those working in the field of PF-PH. My comments are as follows:

In this particular study, the emphasis is on SLUG. Others have shown that mice that overexpress an inducible Slug transgene are viable and appear phenotypically normal; although many of these mice die from cardiac hypertrophy as adults (Perez-Mancera PA, et al *Cytogenet Genome Res.* 2006;114(1):24-29). Also SLUG has been shown in the mouse hypoxia SUGEN model of PH to be associated with the proliferative phenotype of pulmonary arterial smooth muscle cells (PASMCs) and vascular remodelling (Coll-Bonfill et al *PLOS ONE* 11(7): e0159460)

What is the advantage in using the new rat model? Would it have been better to use the inducible SLUG overexpressing mouse in conjunction with bleomycin to investigate and dissect the SLUG/PIP mechanism in PF-PH? I would like to see more in the discussion about the existing mouse literature and how they relate to the proposed mechanism.

2) The authors propose that the SLUG/PIP mechanism is predominantly associated with the alveolar macrophages and PASMCs and based on the evidence provided in Figure 5. The literature suggests that SLUG is expressed in the lung in epithelial cells undergoing EMT. The images in Figures 5B-F are not clear enough to exclude SLUG expression in the epithelial/ mesenchymal compartments. The DAB immunohistochemical staining is difficult to assess in these images because they are small and have various pinkish backgrounds. In addition, I would like to see some immunofluorescence studies of SLUG and alveolar epithelial or fibroblast markers. This point is important because 'predominance' of SLUG expression on macrophages may be due to the inflammatory nature of the MCT/Bleo model rather than a part of PF-PH pathogenesis. I would like to also see better images of staining of SLUG on the patient samples. (Figure 2). Is there any evidence that SLUG expression in the pulmonary epithelium and mesenchymal compartments transient/ associated with EMT?

3) Minor comments: In the supplemental methods, the authors say that 'care was taken not to consider the adventitial fibroblast into consideration during quantification'. How was this done? There is existing evidence that adventitial fibroblasts migrate in the medial and intimal layers of remodelling vessels.

Authors' appeal

14 January 2019

Thank you very much for taking the time to thoroughly review our recent manuscript titled "Histological Hallmarks and Role of Slug/PIP Axis in Pulmonary Hypertension Secondary to Pulmonary Fibrosis (EMM-2018-10061)." We were very encouraged when we were notified that our work was sent to reviewers but were ultimately surprised and disappointed in the final editorial decision. Although reviewers 2 and 3 were more positive and acknowledged the significance of our work, reviewer 1 was less enthusiastic and raised many concerns.

The most important common critique raised by you and the reviewers was about the expression of Slug in different cell types. In the original paper, we showed Slug expression in macrophages; however, we never concluded that macrophages were the only cell type expressing Slug as reviewer 1 implied. On that note, we agree that investigating the expression of Slug in different cell types within the lung is relevant and would add significance to our work. To address this, we started labeling both human and rat lung tissue with Slug antibody co-labeled with markers of smooth muscle cells, endothelial cells, fibroblasts and epithelial cells. Based on our initial experiments, I believe that we are able to show the expression of Slug in different cell types in both rat and human lungs. In addition, we can also show whether Slug expression in different cell types changes between pulmonary fibrosis (PF) and PF-PH as we demonstrated in the original paper for macrophages where the expression of Slug is significantly increased in PF-PH compared to PF. We are also happy to replace some of our IHC images with immunofluorescence and confocal imaging as was mainly suggested by reviewer 1. There are several other points raised by the first reviewer that we can easily clarify as well.

Ultimately, I am writing you to see whether you are willing to consider a revised version of our manuscript. You stated in your decision letter that our work is not acceptable as it stands; but, within 1-2 months, we believe we can improve our paper drastically by including new data suggested by all three reviewers. I sincerely hope that you may reconsider your decision and allow us the opportunity to submit our revised manuscript.

Editor's Response

15 January 2019

Thank you for your email. I understand the different points you raise. Given the potential interest of your findings, we would be willing to consider a new manuscript on the same topic if at some time in the near future you obtained data that would considerably strengthen the message of the study and address the referees concerns in full. To be completely clear, however, I would like to stress that if you were to send a new manuscript, this would be treated as a new submission rather than a revision and would be reviewed afresh, in particular with respect to the literature and the novelty of your findings at the time of resubmission. If you decide to follow this route, please make sure you nevertheless upload a letter of response to the referees' comments.

1st Revision - authors' response

25 May 2019

Response to Reviewer #1:

1. Patient characteristics: There is no data regarding the pulmonary mechanical properties of these patients. How bad was their restrictive disease? Did they had obstructive lung disease as well? How many were on oxygen? We need more details. I strongly suggest providing information on: 1. FVC, FEV1 and FEV1/FVC, 2) DLCO, 3) oxygen saturation, 4) oxygen use.

We agree with this critique and have added all the data suggested (See updated **Table 1**). Since **Table 1** shows the average value \pm SD for each parameter, we have also provided a detailed **supplementary table 1** which gives the individual values of these parameters for each patient. Both PF and PF-PH patients had similar degree of restriction based on forced vital capacity. None of the PF patients and only one PF-PH patients had an obstructive defect based on a FEV1/FVC ratio less than 70%. Virtually all patients were on supplemental oxygen except for one patient in the PF group.

2. These IPF-PH patients seem to have mild PH, the most common form. Yet, there is a subgroup that has PH out of proportion to the underlying fibrosis. This must be carefully pointed out as this sub-type appears to be different (less severe PFTs, very elevated mPAP).

Considering mPAP greater than 40 is one of the most accepted criteria to define severe PH in PF patients, 12 out of 14 of our PF-PH patients (85%) would be classified as mild PH, and 2 patients with mPAP of 44 and 47mmHg, would be classified as severe PH (although there is still no official consensus regarding the definition of mild and severe PH [1]), We chose to keep these two patients in our study since their other parameters did not differ from the remaining PF-PH cohort. For example, the cardiac output of these 2 patients were 4.2 and 3.8L.min⁻¹ (average±SD: 4.7 ± 1); FVC (%predicted) were 52 and 33 (average±SD: 50±20) and FEV1/FVC were 81 and 86 (average±SD: 85 ± 8.4, **Suppl. Table 1**).

3. Figure 1F and G. The vascular pathology seen in severe fibrosis of PF-PH is hard to compare against its corresponding PF alone since it is located within a fibrotic lesions while the PF is located in a less scarred zone. The authors should ensure that they are comparing areas of similar fibrotic burden before quantifying vascular remodeling. This raises concerns regarding the data output on this section of the study.

We agree that comparing areas of similar fibrosis is important for our study. As described in the methods, for all vessels quantified, the fibrosis in the surrounding pulmonary area has been quantified using the Ashcroft score. Then, vessels were grouped in three categories based on the surrounding fibrosis: non-fibrotic area (Ashcroft score 0-2), mild fibrosis (Ashcroft score 2-5) and severe fibrosis (Ashcroft score 5-8). Regarding the representative images, each image shown in the previous version was chosen from the same category of fibrosis, but, considering that these categories are a range, the degree of fibrosis is still variable. For example, in the severe fibrosis category (Ashcroft score between 5-8) an image of severe fibrosis with Ashcroft score of 5 will show noticeably less fibrosis than an image of severe fibrosis with Ashcroft score of 8. Nonetheless, we agree with the reviewer that showing vascular remodeling from images with varying levels of fibrosis raises some concerns; therefore, we have replaced our representative images to show similar scarring for each category (See new **Fig 1F and 1G**).

4. Figure 2A. The Ki67 label is hard to interpret. It looks as if the signal comes mostly from endothelial cells rather than SMC or fibroblasts. I request that the authors do IF and label bot EC and SMC along with ki67 to do colocalization studies. Also, severity of the remodeling is not comparable to that seen in Figure 1.

As suggested, we have removed IHC staining for Ki67 and have performed co-immunofluorescence for Ki67 with vWF, and Ki67 with aSMA to quantify proliferation of endothelial cells and smooth muscle cells, respectively, both in human and rat lungs (See new **Figs. 2A, 5A**). The severity of the remodeling in IF images in **Fig. 2A** seems to be less pronounced than the severity of the remodeling in the Masson Trichrome staining in **Fig 1E, F, G** in PF-PH patients compared to PF since IF only shows SMC or EC layers and not the adventitial layer.

5. Figure 2B. Counter-stain is very faint and makes appreciation the vascular wall anatomy very hard. Please follow my request for IF.

We have removed IHC for Ki67 and performed co-immunofluorescence for Ki67 with SMC and/or EC (new **Fig. 2A, 5A**).

6. Figures 2D-G: It is very hard to appreciate cell number, location and position in Figures 2D and G to be comfortable concerning the analysis. Where is the inset region located in the lung section. There also seems to be much more macrophages in figure E compared to what you see on D and G. Is this only because of the greater magnification? It is hard for me to really accept the quantification based on these poor quality images

We agree with the reviewer that from our original IHC images, it is hard to visualize the expression of Slug in lung tissue. Therefore, we have performed extensive additional co-immunofluorescence experiments to quantify expression of Slug in 5 different cell types both in human and rat lungs from

PF and PF-PH (See new **Figures 2D and 5D**).

7.1. Figure 3. The authors make use of public data to look at genes associated with Slug. A major problem here is that we don't know the clinical characteristics of these patients nor do we know how they compare with their discovery cohort.

We agree with the reviewer that clinical characteristics of our cohort should be comparable with the cohort used in the online microarray database. Although we do not have access to the clinical characteristics of each patient, the average clinical characteristics are reported in the original publication (Table 2 from Mura et al [2]). Overall both cohorts are similar in age, sex, and FVC. Our cohort did have a higher mPAP (32 ± 2 vs 28 ± 5) and a lower DLCO (% predicted: 26.5 ± 9.4 vs. 39 ± 11). We used online microarray data as a discovery tool and the fact that we were able to validate upregulation of prolactin-induced protein (PIP) in our own cohort, further strengthens the implication of PIP in combined PF-PH patients.

7.2 Is slug something that responds differently in the setting of mild vs. severe restriction? Is it independent of IPF severity? This information is mandatory to establish relevance of their discovery to clinical setting.

The restriction in our cohort is not significantly different between PF and PF-PH patients. Our data shows that the percentage of Slug positive macrophages are significantly higher in PF-PH patients. Therefore, we plotted the percentage of Slug positive macrophages as a function of FVC for each PF (**A**) and PF-PH patients (**B**). Although we did not find any significant association between the expression of Slug and restriction severity in PF nor in PF-PH, a trend toward an association can be seen in PF-PH. However, our limited number of patients for this type of investigation prevents us from drawing any conclusion. We also did not find any significant association between Slug mRNA and FVC in our PF and PF-PH patients (**C**). Due to the small number of patients for which we had access to lung samples for mRNA measurements, we pulled together PF and PF-PH patients to examine a potential correlation between Slug mRNA and FVC.

8. Figure 3D-F: IHC for PIP appears to be all over the PF lungs (PH or not). Why are we looking only at SMCs? The ECs and fibroblasts are also part of the path-biology of vascular remodeling in this setting as they are vital components of the vessel wall. Please perform proliferation/apoptosis studies using all 3 cell types. It would also add greater impact to your paper if you have PBMCs from IPF patients to look at the Slug/PIP production via FACS and in co-culture assays.

We agree that PIP is an extracellular matrix protein, thus could have an effect on multiple cell types

in the lung. We exposed PAECs, PASMCs and fibroblasts to different concentrations of PIP and measured proliferation for each cell type. The results of these experiments demonstrate a significant increase in PAEC and PASMC proliferation when exposed to PIP, while PIP had no effect on the fibroblasts (new **Fig 3G-J**). We also agree that showing Slug/PIP production in PBMC from PF and PF-PH patients would have been a great addition to our paper. However, in our preliminary experiments, we were not able to detect Slug transcripts in PBMC from PF and PF-PH patients.

9. Figure 5. Once again, the IHC is very problematic. I zero in on the IF because it allows me to appreciate the slug signal better (macrophages are notorious for nonspecific stains in IHC) but I can't rule out that slug is not being expressed in other parts of the lung. Are we POSITIVE that slug is only being expressed in macrophages?

In our original submission, we showed Slug expression in macrophages; however, we never concluded that macrophages were the only cell type expressing Slug. We agree with the reviewer that it is important to examine which cell types express Slug in the lungs. Therefore, we performed coimmunofluorescence and measured expression of Slug in 5 different cell types (epithelial cells, macrophages, fibroblasts, pulmonary arterial smooth muscle cells and endothelial cells) in human and rats. These new experiments revealed that Slug is expressed in epithelial cells, macrophages and fibroblasts. Interestingly, expression of Slug was significantly higher only in macrophages in PF-PH compared to PF both in rat lungs similar to humans (See new **Fig. 2D and 5D**). Since expression of Slug was only different in macrophages, we next examined expression of Slug in fibrotic vs. nonfibrotic areas. We found that the percentage of Slug-positive macrophages is increased significantly both in non-fibrotic and fibrotic areas of the lung in PF-PH compared to PF patients and rats (See new **Fig. 2F and 5F**).

10. Figure 6 and 7. The Slug siRNA should affect many cells since it is not targeted to macrophages. This leads me to ask again: are we positive that slug is only located to macrophages? The authors make no effort to characterize whether slug expression is down in the lungs using qPCR/WB or to purify macrophages and measure slug/PIP production after treatment. The remodeled vessels do not look very different to my eye which again raises concerns regarding the quantification used.

In the original submission, we showed Slug and PIP transcripts (using qPCR) were significantly downregulated in the lung of rats treated with Si-Slug compared to Si-Scrm. As suggested, we have performed Western Blot analysis for Slug and PIP. Our new data shows Slug and PIP proteins are significantly downregulated in the lungs of rats that received Si-Slug compared to Si-Scrm RNA (See New **Fig.7B-C and Fig 8D-E**). We also show the percentage of Slug positive macrophages is decreased in the lungs of rats treated with Si-Slug vs. Si-Scrm (**Fig 8A and 8C**).

Regarding vascular remodeling, we carefully chose representative images that reflect our quantification, which is a ~20% decrease in vascular remodeling in Si-Slug compared to Si-Scrm treated rats. We replaced our images to better show this decrease in the vascular remodeling. For quantification of vascular remodeling, we would like to clarify that we only took the media and intima of the vessels. The adventitia was not taken into account in our measurement since it lacks a clear anatomical delineation for most vessels which makes the quantification of the adventitial thickening difficult [3,4].

Response to reviewer #2:

Figure 1 B, C and D, Include the expression levels in control and PAH will give more strength to the results if they match with the results in figure 4 I, J and K.

We agree with the reviewer that these data from Ctrl and PAH patients would be of interest. Unfortunately, for Ctrl and PAH patients, we only had access to slides and did not have any lung samples to measure transcript and protein expression. For the PF and PF-PH patients, out of 14 patients in each group, we had access to the slides and lung samples from 6 in each group, and for remaining 8, we only had access to slides.

Figure 2 and 3 D: Include Western blot analysis for Slug and PIP in lung homogenate will be more informative.

As suggested, we have added Western blots for Slug and PIP for human and rat lungs. Our new

Western Blot analysis show protein expression of Slug and PIP are significantly increased in human and rat lungs confirming our mRNA finding (See new **Fig. 2C, 3E, 5C, 6B**). We also found a significant decrease of Slug and PIP protein expression in rats treated with Si-Slug compared to Si-Scrm treated rats (seen new **Fig 7B and 8E**)

Figure 4: Panel B,C,D the authors are showing only 4 rats after Bleo treatment vs 7 controls while in the other set (panels F and G) the number is the same. I would like to know the main reason to use a less sample number and ask for the increase of the n after bleo treatment to have the same number.

In the previous submission, we had only 4 rats with 2 weeks of bleomycin. The aim of this experiment was only to confirm the presence of lung fibrosis after two weeks of bleomycin (which has been reported previously in numerous publications [5- 8]) before exposure to the second insult of MCT for triggering PH. Nonetheless, we agree with the reviewer that the number of animals should be similar to the other groups, so we have increased the number of rats to 7 in this group. We have updated our measurements of Ashcroft score, collagen I, collagen III, and fibronectin using 7 rats in each group (See new **Fig 4A-D**).

Since there are post-transcriptional regulation mechanisms, it would be important to show the protein levels of fibronectin and collagen (panel I, J, K)

We agree that ECM proteins are subjected to post-transcriptional regulation, which could impact the protein stability without modifying mRNA expression. However, our Ashcroft score measurements from Masson's trichrome staining, which allows visualization and quantification of the global ECM protein deposition, did not reveal significant differences in the extent of ECM protein deposition between PF and PF-PH, supporting our mRNA quantification. Our Ashcroft score measurements are supported by previous work showing no significant differences between lung fibrosis in PF-PH patients compared to PF [9- 11]. Therefore, based on all these evidence from molecular and clinical investigations, we do not believe there is a significant difference in post-transcriptional modification of these ECM proteins between PF-PH and PF patients in our cohort.

The authors are not showing enough data to prove the molecular mechanism through Slug via PIP is regulating PSMCs proliferation; since Slug has more targets that could play a role during PF-PH vascular remodeling;

Our in-vivo experiments showed Slug inhibition using Si-Slug was associated with decreased expression of PIP and decreased vascular remodeling (new **Fig. 7J and 8D-E**). In agreement with our in-vivo experiments, our in-vitro experiments on PSMC, PAEC and fibroblasts, showed exogenous PIP is able to induce proliferation both in PSMC and PAEC (**Fig. 3G-J**). These experiments together support the role of Slug/PIP axis in promoting pulmonary vascular remodeling. Our working hypothesis is that increased expression of Slug in macrophages in PF-PH is associated with up regulation of PIP in the extracellular matrix, which in turn promotes proliferation of PSMC and PAEC leading to pulmonary vascular remodeling.

Our microarray analysis of PF-PH patients compared to PF revealed seven genes that are transcriptional targets of Slug and are implicated in both extracellular matrix and cellular proliferation. Since the most up-regulated gene, Mucin 4, is anchored to the cell membrane [12] and is not secreted into the extracellular matrix, we focused on the second most up-regulated transcriptional target of Slug, which is PIP. As pointed out by the reviewer, it is still possible that Slug promotes vascular remodeling in PF-PH by controlling other extracellular proteins than PIP. This possibility is discussed in the manuscript.

I recommend the isolation of PSMCs cells after Bleo, MCT and siSlug with the respective controls and perform BrdU experiment and the measurement of PIP mRNA expression and the protein levels of Slug and PIP via western blot analysis.

After performing co-localization studies with Slug and aSMA antibodies, we found that Slug is not expressed in PSMC in human and rats lungs (new **Fig. 2D and Fig. 5D**). Since PIP is an extracellular matrix protein, we did not think that further investigation into the Slug/PIP axis within PSMCs is helpful.

Which pathways they suggest are increasing the vascular remodeling in PF-PH? Vascular remodeling is not only due to the increase in cell proliferation, it is also due to the production of ECM components, however the author didn't find any change in the expression of CollI, ColIII and fibronectin, may be interesting analyze the expression levels of other ECM proteins.

We agree with the reviewer that ECM modification is part of the vascular remodeling seen in pulmonary hypertension. In group 3 PH Milara et al. [13] already demonstrated the modification of the vascular ECM using isolated pulmonary vessels. In the present study we did not find significant differences in the expression of collagens and fibronectin between PF and PF-PH. In our view, the lack of a change should not be considered as the absence of increased deposition of vascular ECM. These proteins are also part of the lung fibrosis so local vascular changes of the ECM may be masked due to the high concentration of these proteins in the lung parenchyma. This section has been added to the discussion.

Page 13 line 2 the data are not enough to assert the role of ECM in cellular communication between macrophages and vascular cells in PF-PH, the expression of ECM proteins doesn't change and the results show an increase in wall thickening in non-fibrotic and fibrotic areas.

We agree with the reviewer and have removed that sentence.

Figure 8: The authors found an increase in the expression of Slug in macrophages, however; in previous reports was prove that also epithelial cells have a high expression of Slug in PF. Which are the expression levels of Slug in PF-PH vs PF? The expression of Slug in epithelial cells could have the same effect over PIP and proliferation in PASCAM.

We agree with the reviewer comment and performed co-immunofluorescence to quantify Slug expression in 5 major cell types in the lungs (epithelial cells, macrophages, endothelial cells, smooth muscle cells, and fibroblasts). Our new data shows Slug is highly expressed in the epithelial cells and to lesser extent in macrophages, fibroblasts and endothelial cells (new **Fig. 2D and 5D**). More importantly, we found that expression of Slug is significantly increased ONLY in macrophages in PF PH compared to PF both in the lungs of humans and rats, and not in other cell types. Upregulation of Slug in macrophages in PF-PH is associated with increase PIP expression, therefore, the increase in PIP in PF-PH is mainly mediated by macrophages. While Slug is highly expressed in epithelial cells, it is unlikely that epithelial cells are participating in upregulation of PIP in PF-PH since there is no change in Slug expression within epithelial cells between PF and PF-PH groups.

Figure 7: (B) and (C) are interchanged in the foot note. D) Quantification of Slug..... But in the plot is write PIP

We have corrected the footnote.

Response to Reviewer #3

What is the advantage in using the new rat model? Would it have been better to use the inducible SLUG over-expressing mouse in conjunction with bleomycin to investigate and dissect the SLUG/PIP mechanism in PF-PH? I would like to see more in the discussion about the existing mouse literature and how they relate to the proposed mechanism.

We developed our combined model of PF-PH in rats, since it is well accepted that rats mimic histopathological features of human PH better than mice. Mice never develop severe pulmonary vascular remodeling as it is evident by their much lower RVSP (~35 mmHg vs 60-80 mmHg in rats [14]). Bleomycin has been used for decades to induce PF; however, we found that rats treated with bleomycin alone do not recapitulate the histological and molecular features of PF-PH patients. For example, bleomycin alone only induces vascular remodeling within the fibrotic areas, and Slug is not up-regulated in the lungs of rats in bleomycin alone group (**Figure 4M and 5C** and [15]). It took us 4 years to develop a novel translational rat model of combined PF-PH that is reproducible and shares similar histological (fibrosis, pulmonary vascular remodeling) and molecular features (Slug and PIP) with human PF-PH. We agree with the reviewer that slug inducible macrophage specific mouse would be a great tool to further examine the role of Slug in promoting PF-PH transition, but

at the moment there is no macrophage or monocyte specific promoter [16,17]. Regarding mice literature in the context of PF-PH, as suggested by the reviewer, we found 2 studies that shows mice on chronic low dose of bleomycin also develop some degree of PH [18,19]. The development of PH in mice on chronic dose of bleomycin was mitigated by hyaluronan synthase inhibition [18] and also in smooth muscle cell Adenosine A2b Receptor knock out mice [19]. We have added these studies to our discussion. However, these studies did not investigate the role of SLUG/PIP axis in development of PH in preexisting PF.

2) The authors propose that the SLUG/PIP mechanism is predominantly associated with the alveolar macrophages and PASMCs and based on the evidence provided in Figure 5. The literature suggests that SLUG is expressed in the lung in epithelial cells undergoing EMT. The images in Figures 5B-F are not clear enough to exclude SLUG expression in the epithelial/ mesenchymal compartments. The DAB immunohistochemical staining is difficult to assess in these images because they are small and have various pinkish backgrounds. In addition, I would like to see some immunofluorescence studies of SLUG and alveolar epithelial or fibroblast markers. This point is important because 'predominance' of SLUG expression on macrophages may be due to the inflammatory nature of the MCT/Bleo model rather than a part of PF-PH pathogenesis. I would like to also see better images of staining of SLUG on the patient samples. (Figure 2). Is there any evidence that SLUG expression in the pulmonary epithelium and mesenchymal compartments transient/ associated with EMT?

According to the reviewer's comment, we measured the expression of Slug in human and rat lungs in 5 different cell types (epithelial cells, macrophages, fibroblast, endothelial cells and smooth muscle cells) (new **Fig. 2D and 5D**). We agree with the reviewer that the inflammatory nature of MCT could be a confounding factor. Nonetheless, qPCR studies performed on human lungs and lungs of our rat model do not demonstrate any significant difference in CD68 transcript levels between PF and PF-PH groups (**Fig. 2E and 5E**). These data suggest that the sequential administration of bleomycin and MCT does not induce an increased recruitment of macrophages compared to the bleomycin alone in our study. In our new quantification of Slug expression, we detected expression of Slug in epithelial cells as well as in fibroblasts, but the percentage of Slug positive cells did not differ in epithelial cells and fibroblasts between PF and PF-PH patients and rats. Examining role of Slug in promoting EMT in combined PFPH model is the focus of our ongoing research but is beyond the scope of this paper.

3) Minor comments: In the supplemental methods, the authors say that 'care was taken not to consider the adventitial fibroblast into consideration during quantification'. How was this done? There is existing evidence that adventitial fibroblast migrate in the medial and intimal layers of remodeling vessels.

To distinguish the fibroblasts in the previous quantification by IHC we used the external laminae to delimitate the media. As pointed out by the reviewer, we could not distinguish fibroblast migrating into the media or the intima of the vessels in IHC, but we did not take into account the adventitial fibroblasts. In our new quantification by immuno-fluorescence, only the vWF or aSMA positive cells were taken into account. Nonetheless, fibroblasts can also express aSMA and so this is a limitation of our new immuno-fluorescence quantifications of SMC since aSMA positive fibroblasts will be considered in this quantification.

References:

- [1] Trammell AW, Pugh ME, Newman JH, Hemnes AR, Robbins IM. Use of Pulmonary Arterial Hypertension–Approved Therapy in the Treatment of Non–Group 1 Pulmonary Hypertension at US Referral Centers. *Pulm Circ* 2015;5:356–363. doi:10.1086/681264.
- [2] Mura M, Anraku M, Yun Z, McRae K, Liu M, Waddell TK, et al. Gene expression profiling in the lungs of patients with pulmonary hypertension associated with pulmonary fibrosis. *Chest* 2012;141:661–673. doi:10.1378/chest.11-0449.
- [3] Pietra GG, Capron F, Stewart S, Leone O, Humbert M, Robbins IM, et al. Pathologic assessment of vasculopathies in pulmonary hypertension. *J Am Coll Cardiol* 2004;43:S25–S32. doi:10.1016/j.jacc.2004.02.033.
- [4] Dorfmueller P. Pulmonary hypertension: pathology. *Handb Exp Pharmacol* 2013;218:59–75. doi:10.1007/978-3-642-38664-0_3.
- [5] Chaudhary NI, Schnapp A, Park JE. Pharmacologic differentiation of inflammation and fibrosis in the rat bleomycin model. *Am J Respir Crit Care Med* 2006;173:769–776.

doi:10.1164/rccm.200505-717OC.

[6] Stellari FF, Ruscitti F, Pompilio D, Ravanetti F, Tebaldi G, Macchi F, et al. Heterologous Matrix Metalloproteinase Gene Promoter Activity Allows In Vivo Real-time Imaging of Bleomycin-Induced Lung Fibrosis in Transiently Transgenized Mice. *Front Immunol* 2017;8.

doi:10.3389/fimmu.2017.00199.

[7] Chen K-J, Li Q, Weng C-M, Duan Z-X, Zhang D-D, Chen Z-Q, et al. Bleomycin-enhanced alternative splicing of fibroblast growth factor receptor 2 induces epithelial to mesenchymal transition in lung fibrosis. *Biosci Rep* 2018;38:BSR20180445. doi:10.1042/BSR20180445.

[8] Lei Y, Wang K, Li X, Li Y, Feng X, Zhou J, et al. Cell-surface translocation of annexin A2 contributes to bleomycin-induced pulmonary fibrosis by mediating inflammatory response in mice. *Clin Sci* 2019;133:789–804. doi:10.1042/CS20180687.

[9] Zisman DA, Karlamangla AS, Ross DJ, Keane MP, Belperio JA, Saggari R, et al. High-resolution chest CT findings do not predict the presence of pulmonary hypertension in advanced idiopathic pulmonary fibrosis. *Chest* 2007;132:773–779. doi:10.1378/chest.07-0116.

[10] Nadrous HF, Pellikka PA, Krowka MJ, Swanson KL, Chaowalit N, Decker PA, et al. Pulmonary hypertension in patients with idiopathic pulmonary fibrosis. *Chest* 2005;128:2393–2399. doi:10.1378/chest.128.4.2393.

[11] Nathan SD, Barbera JA, Gaine SP, Harari S, Martinez FJ, Olschewski H, et al. Pulmonary hypertension in chronic lung disease and hypoxia. *Eur Respir J* 2019;53.

doi:10.1183/13993003.01914-2018.

[12] Dhanisha SS, Guruvayoorappan C, Drishya S, Abeesh P. Mucins: Structural diversity, biosynthesis, its role in pathogenesis and as possible therapeutic targets. *Crit Rev Oncol Hematol* 2018;122:98–122. doi:10.1016/j.critrevonc.2017.12.006.

[13] Milara J, Escrivá J, Ortiz JL, Juan G, Artigues E, Morcillo E, et al. Vascular effects of sildenafil in patients with pulmonary fibrosis and pulmonary hypertension: an ex vivo/in vitro study. *Eur Respir J* 2016;47:1737–1749. doi:10.1183/13993003.01259-2015.

[14] Vitali SH, Hansmann G, Rose C, Fernandez-Gonzalez A, Scheid A, Mitsialis SA, et al. The Sugen 5416/hypoxia mouse model of pulmonary hypertension revisited: long-term follow-up. *Pulm Circ* 2014;4:619–629. doi:10.1086/678508.

[15] Jayachandran A, Königshoff M, Yu H, Rupniewska E, Hecker M, Klepetko W, et al. SNAI transcription factors mediate epithelial-mesenchymal transition in lung fibrosis. *Thorax* 2009;64:1053–1061. doi:10.1136/thx.2009.121798.

[16] Shi J, Hua L, Harmer D, Li P, Ren G. Cre Driver Mice Targeting Macrophages. *Methods Mol Biol Clifton NJ* 2018;1784:263–275. doi:10.1007/978-1-4939-7837-3_24.

[17] Hume DA. Applications of myeloid-specific promoters in transgenic mice support in vivo imaging and functional genomics but do not support the concept of distinct macrophage and dendritic cell lineages or roles in immunity. *J Leukoc Biol* 2011;89:525–538.

doi:10.1189/jlb.0810472.

[18] Collum SD, Chen N-Y, Hernandez AM, Hanmandlu A, Sweeney H, Mertens TCJ, et al. Inhibition of hyaluronan synthesis attenuates pulmonary hypertension associated with lung fibrosis. *Br J Pharmacol* 2017;174:3284–3301. doi:10.1111/bph.13947.

[19] Mertens TCJ, Hanmandlu A, Tu L, Phan C, Collum SD, Chen N-Y, et al. Switching-Off Adora2b in Vascular Smooth Muscle Cells Halts the Development of Pulmonary Hypertension. *Front Physiol* 2018;9:555. doi:10.3389/fphys.2018.00555.

2nd Editorial Decision

30 June 2019

Thank you for the submission of your revised manuscript to EMBO Molecular Medicine. As you will see the reviewers are now globally supportive and I am pleased to inform you that we will be able to accept your manuscript pending the following final amendments:

1) Please address referee 3's comment in writing. At this stage, we'd like you to discuss this referee's concern and if you do have data at hand, we'd be happy for you to include it, however we will not ask you to provide any additional experiments.

I look forward to receiving your revised manuscript.

***** Reviewer's comments *****

Referee #1 (Remarks for Author):

My concerns from the original submission have been addressed.

Referee #2:

Suitable for publication.

Referee #3 (Remarks for Author):

On the whole the authors have addressed the majority of the reviewers' comments satisfactorily.

COMMENT:

I am of the view that the authors should be using macrophage inducible targeting strategy to assess the role of Slug in pulmonary hypertension secondary to pulmonary fibrosis and understand the role of the pulmonary macrophage subsets. Despite the authors' comment, the literature supports that there are promoters that specifically target interstitial macrophages (for example CX3CR1-ERCre as described in McCubbrey et al. 2017: Promoter specificity and efficacy in conditional and inducible transgenic targeting of lung macrophages' *Front Immunol*; 8:1618). To strengthen their findings, they could even deplete blood derived monocyte/macrophages using Clodronate in the rat model.

2nd Revision - authors' response

31 July 2019

Referee #1 (Remarks for Author):

My concerns from the original submission have been addressed.

Referee #2:

Suitable for publication.

Referee #3 (Remarks for Author):

On the whole the authors have addressed the majority of the reviewers' comments satisfactorily.

COMMENT:

I am of the view that the authors should be using macrophage inducible targeting strategy to assess the role of Slug in pulmonary hypertension secondary to pulmonary fibrosis and understand the role of the pulmonary macrophage subsets. Despite the authors' comment, the literature supports that there are promoters that specifically target interstitial macrophages (for example CX3CR1-ERCre as described in McCubbrey et al. 2017: Promoter specificity and efficacy in conditional and inducible transgenic targeting of lung macrophages' *Front Immunol*; 8:1618). To strengthen their findings, they could even deplete blood derived monocyte/macrophages using Clodronate in the rat model.

We agree with the reviewer that using a slug-inducible macrophage-specific mouse would be a great tool to further examine the role of Slug in macrophages in promoting PF-PH transition using CX3CR1-ER Cre mice. However, since mouse models of PH do not develop as severe pulmonary vascular remodeling as rats and do not mimic the histopathological features of human PH, we developed our combined model of PF-PH in rats. In rats the use of Clodronate would have been an interesting option to investigate the role of macrophages in PH development secondary to PF. Nonetheless, this approach also has limitation, since Clodronate has already been shown to reduce pulmonary fibrosis during the fibrotic phase of bleomycin model (Gibbons *et al*, 2011). In addition, macrophages depletion has already been extensively demonstrated to decrease pulmonary hypertension in different animal models (Tian *et al*, 2013; Thenappan *et al*, 2011; Rabinovitch *et al*, 2014; Žaloudíková *et al*, 2016). Thus, we believe our best strategy was to use si-RNA targeting Slug to inhibit Slug expression in the lungs of PF-PH rats.

References:

Gibbons MA, MacKinnon AC, Ramachandran P, Dhaliwal K, Duffin R, Phythian-Adams AT, van Rooijen N, Haslett C, Howie SE, Simpson AJ, Hirani N, Gaudie J, Iredale JP, Sethi T & Forbes SJ

(2011) Ly6Chi monocytes direct alternatively activated profibrotic macrophage regulation of lung fibrosis. *Am. J. Respir. Crit. Care Med.* **184**: 569–581

Rabinovitch M, Guignabert C, Humbert M & Nicolls MR (2014) Inflammation and immunity in the pathogenesis of pulmonary arterial hypertension. *Circ. Res.* **115**: 165–175

Thenappan T, Goel A, Marsboom G, Fang Y-H, Toth PT, Zhang HJ, Kajimoto H, Hong Z, Paul J, Wietholt C, Pogoriler J, Piao L, Rehman J & Archer SL (2011) A central role for CD68(+) macrophages in hepatopulmonary syndrome. Reversal by macrophage depletion. *Am. J. Respir. Crit. Care Med.* **183**: 1080–1091

Tian W, Jiang X, Tamosiuniene R, Sung YK, Qian J, Dhillon G, Gera L, Farkas L, Rabinovitch M, Zamanian RT, Inayathullah M, Fridlib M, Rajadas J, Peters-Golden M, Voelkel NF & Nicolls MR (2013) Blocking macrophage leukotriene b4 prevents endothelial injury and reverses pulmonary hypertension. *Sci. Transl. Med.* **5**: 200ra117

Žaloudíková M, Vytášek R, Vajnerová O, Hniličková O, Vízek M, Hampl V & Herget J (2016) Depletion of alveolar macrophages attenuates hypoxic pulmonary hypertension but not hypoxia-induced increase in serum concentration of MCP-1. *Physiol. Res.* **65**: 763–768

Corresponding Author Name: Mansoureh Eghbali

Journal Submitted to: EMBO Mol. Med.

Manuscript Number: EMM-2018-10061